Subject Area:
developmental biology

Keywords:
FAK, Wnt, gastrulation, cell migration, convergent extension, actin

Authors for correspondence:
Tang-Long Shen
e-mail: shentl@ntu.edu.tw
Shyh-Jye Lee
e-mail: jefflee@ntu.edu.tw

# Wnt5b integrates Fak1a to mediate gastrulation cell movements via Rac1 and Cdc42

I-Chen Hung[1], Tsung-Ming Chen[1,2,3], Jing-Ping Lin[2], Yu-Ling Tai[2], Tang-Long Shen[2,5] and Shyh-Jye Lee[1,4,5,6]

[1]Department of Life Science, and [2]Department of Plant Pathology and Microbiology, National Taiwan University, No. 1, Roosevelt Road, Section 4, Taipei 10617, Taiwan
[3]Department and Graduate Institute of Aquaculture, National Kaohsiung Marine University, Kaohsiung, Taiwan
[4]Research Center for Developmental Biology and Regenerative Medicine, [5]Center for Biotechnology, and [6]Center for Systems Biology, National Taiwan University, Taipei, Taiwan

I-CH, 0000-0003-2584-8457; T-MC, 0000-0002-5462-1668; J-PL, 0000-0003-0686-0215; T-LS, 0000-0001-6264-3608; S-JL, 0000-0001-8452-9152

Focal adhesion kinase (FAK) mediates vital cellular pathways during development. Despite its necessity, how FAK regulates and integrates with other signals during early embryogenesis remains poorly understood. We found that the loss of Fak1a impaired epiboly, convergent extension and hypoblast cell migration in zebrafish embryos. We also observed a clear disturbance in cortical actin at the blastoderm margin and distribution of yolk syncytial nuclei. In addition, we investigated a possible link between Fak1a and a well-known gastrulation regulator, Wnt5b, and revealed that the overexpression of *fak1a* or *wnt5b* could cross-rescue convergence defects induced by a *wnt5b* or *fak1a* antisense morpholino (MO), respectively. Wnt5b and Fak1a were shown to converge in regulating Rac1 and Cdc42, which could synergistically rescue *wnt5b* and *fak1a* morphant phenotypes. Furthermore, we generated several alleles of *fak1a* mutants using CRISPR/Cas9, but those mutants only revealed mild gastrulation defects. However, injection of a subthreshold level of the *wnt5b* MO induced severe gastrulation defects in *fak1a* mutants, which suggested that the upregulated expression of *wnt5b* might complement the loss of Fak1a. Collectively, we demonstrated that a functional interaction between Wnt and FAK signalling mediates gastrulation cell movements via the possible regulation of Rac1 and Cdc42 and subsequent actin dynamics.

## 1. Introduction

Vertebrate gastrulation is vital to establish germ layers and body axes by coordinated cell movements in zebrafish [1,2]. At mid-gastrulation, the prospective mesendoderm cells are internalized at the blastoderm margin to form the hypoblast [3,4]. Hypoblast cells migrate on the dorsal epiblast via tightly regulated cell–cell and cell–extracellular matrix (ECM) adhesions and move anteriorly to become mesendodermal layers. Meanwhile, ventral and lateral cells converge and extend anterior-dorsally to elongate and narrow the germ layers to establish the anterior–posterior body axis [5].

Focal adhesion kinase (FAK) is a non-receptor tyrosine kinase and is central to the regulation of cell movements and cell–ECM adhesions [6,7]. It is autophosphorylated upon activation by integrin, growth factor stimuli and/or G-protein-coupled signalling [8]. The phosphorylated FAK binds to activated Src to phosphorylate additional tyrosine residues on FAK and then recruits other proteins to modulate distinct signal transduction pathways involved in regulating multiple cellular functions such as cell adhesion, spreading and migration via cytoskeletal reorganization [9–11].

FAK is evolutionarily conserved in mammals and lower eukaryotic organisms [12]. It is highly expressed during embryogenesis in zebrafish [13], African clawed frog [14], chicken [15] and mouse [16]. FAK has a vital role in embryogenesis [17]. *FAK*-null mice die at embryonic day 8.5 with defects in the axial mesoderm and cardiovascular system [18,19]. These abnormalities are similar to those of fibronectin- and integrin-knockout mice that may be due to a disorganized cytoskeleton [20], impaired cellular migration and adhesion [21], or cessation of mesodermal cell proliferation during embryogenesis [22]. FAK is critical for epiboly, cell polarity and intercalation during embryogenesis in *Xenopus laevis* [23–25]. Other cell signalling pathways such as Wnt signalling are also involved in gastrulation [26]. How those different signals are coordinated during gastrulation remains unclear.

The non-canonical Wnt pathway, which mediates planar cell polarity (PCP) via Frizzled or the tyrosine kinase (Ryk)-related receptor, is a permissive cue for cell migration during development [27,28]. Non-canonical Wnt, *wnt11* and *wnt5b* mutants are defective in cell migration during gastrulation in zebrafish [29,30]. Knockdown of *wnt5a* expression reduces focal adhesion dynamics by affecting FAK phosphorylation in cellular assays [31]. However, zebrafish *wnt5a* messenger RNA (mRNA) gradually decreases within 4 h post-fertilization (hpf). By contrast, another isoform of *wnt5*, *wnt5b*, is highly expressed at the margin of the epiblast during epiboly when *fak* is also expressed [32,33]. This implies a possible interaction between Fak and Wnt5b during gastrulation.

Two FAK paralogous genes, *fak1a* and *fak1b*, are identified in the zebrafish genome [34]. A comparative study showed that the *fak1b* locus is a duplicate of *fak1a* locus and both of them share syntenies with the site of the human FAK locus. High conservation of many important protein domains and 69% identity through the peptide sequence indicates partially overlapping functions of *fak1a* and *fak1b* [34]. Zebrafish Fak1a and Fak1b are highly similar to mammalian FAK (electronic supplementary material, figure S1). Using antisense morpholino (MO) oligonucleotides, we observed Fak1a MO caused severe gastrulation phenotypes than that of *fak1b* MO in zebrafish embryos. Primary amino acid sequence and phylogenetic analyses also showed Fak1a rather than Fak1b is more related to human FAK (electronic supplementary material, figure S2); thus, here we primarily focus on the role of Fak1a and its interaction with Wnt5b during gastrulation.

In this study, we show that the loss of Fak1a impairs gastrulation cell movements via regulation of actin dynamics in zebrafish. Fak1a was found to collaborate with Wnt5b in controlling Rac1 and Cdc42 activities to mediate gastrulation cell movements. We generated several *fak1a* zebrafish mutant alleles; however, those alleles only exhibited mild gastrulation defects. Interestingly, a subthreshold level of the *wnt5b* MO could induce a more severe gastrulation phenotype in the *fak1a* mutants. This suggests Wnt5b may compensate the loss of Fak1a in the mutants.

# 2. Material and methods

## 2.1. Cell culture and transfection

293T and FAK-null mouse embryonic fibroblast ($FAK^{-/-}$) cells were maintained in a 5% $CO_2$ incubator at 37°C in Dulbecco's modified Eagle's medium (DMEM) (Hyclone Laboratories, Logan, UT, USA) containing 10% fetal bovine serum (FBS) or 10% calf serum (Hyclone Laboratories), respectively. All cells were transfected with the desired reagent using Lipofectamine 2000™ (Invitrogen Life Technology, Carlsbad, CA, USA) according to the manufacturer's instructions.

## 2.2. Zebrafish fak cloning and expression vector construction

Coding sequences (CDSs) of *fak1a* and *fak1b* were amplified from total zebrafish complementary (c)DNAs by a reverse transcription–polymerase chain reaction (RT–PCR). Total RNAs of zebrafish were isolated by Trizol (Invitrogen) and reverse-transcribed using MMLV reverse transcriptase (Promega, Madison, WI, USA). Primers were designed according to reference RNA sequences (NM_131796.1 and NM_198819.1). The PCR was carried out at 95°C for 5 min followed by amplification at 95°C for 30 s; 58°C for 30 s; and 72°C for 30 s for 30 cycles with the proper primer set (electronic supplementary material, table S1). PCR products were cloned into the pGEM-T vector (Promega), sequenced and analysed. The correct *fak1a* CDS was subcloned into pEGFP, pKH3 or pCS2+ vectors for the overexpression experiments.

## 2.3. Immunofluorescence cell staining

Cells grown on a sterile cover glass were fixed in 4% paraformaldehyde (PFA) for 15 min. The cover glass was thoroughly washed with phosphate-buffered saline (PBS), blocked with 10% FBS and 0.3% Triton X-100, and then incubated overnight at 4°C with different primary antibodies against FAK, human influenza hemagglutinin (HA) and enhanced green fluorescent protein (EGFP; Santa Cruz Biotechnology, Santa Cruz, CA, USA). Samples were incubated with their corresponding secondary antibody and then stained by Hoechst 33258 dye (Sigma-Aldrich, St Louis, MO, USA) to reveal nuclei. Fluorescence signals were recorded with an epifluorescence microscope (Model IX71, Olympus, Japan) and analysed using Image-Pro Plus software (Media Cybernetics, Silver Spring, MD, USA).

## 2.4. 5-bromodeoxyuridine (BrdU) incorporation assays

Cells were serum-starved for 24 h in DMEM with 0.5% FBS to arrest them in G0 phase and then incubated for 24 h in DMEM containing 10% serum and 150 µM BrdU (Sigma). Cells were then washed with PBS, fixed in 4% PFA, permeabilized with 0.3% Triton X-100, digested with DNase I (New England Biolabs, Ipswich, MA, USA), blocked with 10% FBS, and processed for immunofluorescent staining with anti-BrdU (1 : 200, Sigma) and anti-GFP (1 : 200) antibodies, as described previously [35]. Cells counted in multiple fields were scored for BrdU-positive cells in three independent experiments.

## 2.5. Cell migration assay

Cell migration was measured in a 48-well chemotaxis Boyden chamber using 8 µm polyvinyl pyrrolidone-free polycarbonate membranes as described previously [18]. The membrane was placed over the bottom chamber, and cells were loaded in the upper well of a chamber. After 6 h, the membrane was stained with crystal violet. Finally, the number of stained cells was

royalsocietypublishing.org/journal/rsob   *Open Biol.* **10**: 190273

calculated from five randomly selected fields of each well under a microscope at 20× magnification.

## 2.6. Cell transplantation

For transplantation preparation, donor embryos were injected with 1% rhodamine dextran and 5 ng of standard control MO (StdMO) or *fak1a* tMO$_1$, respectively. Cells from donor embryos were transplanted into untreated embryos and *fak1a* morphants separately at sphere stage as described previously [36]. Transplanted cells were further monitored and recorded by using a Leica DM5000 B microscope with a charge-coupled device (CCD) camera.

## 2.7. Immunoblotting

Cells were lysed in RIPA lysis buffer, and the protein concentration was determined by the Bradford assay (Bio-Rad, Hercules, CA, USA). Whole-cell lysates were mixed with an equal amount of 4× sodium dodecylsulfate (SDS) sample buffer, boiled and subjected to SDS–polyacrylamide gel electrophoresis (PAGE). Proteins were subsequently transferred onto a polyvinylidene difluoride membrane (Amersham, Buckinghamshire, UK) and subjected to immunostaining against designated antibodies. Antibodies against FAK (C-20) and actin were purchased from Santa Cruz Biotechnology. The α-pY397-FAK, α-pY861-FAK, α-pY576/577-FAK, paxillin and α-pY118-paxillin antibodies were from BD Transduction Laboratories (Palo Alto, CA, USA). The polyclonal α-extracellular signal-regulated kinase (ERK)1/2, α-pERK (pT202/pY204), α-p38 MAPK, α-phospho-p38 MAPK (Thr-180/Tyr-182), α-AKT and α-pS473-AKT antibodies were purchased from Cell Signaling (Danvers, MA, USA). The α-phosphate tyrosine (pY20, clone 4G10; Millipore, Bedford, MA, USA) and α-S732 phosphorylated FAK (Sigma) antibodies were also used in this study. Stained protein bands were detected using a Western-Light ECL kit (PerkinElmer, Waltham, MA, USA).

## 2.8. Zebrafish maintenance and embryo culture

AB zebrafish were maintained at 28.5°C on a 14 h light/10 h dark cycle. Embryos collected from natural mating were maintained and staged according to Kimmel *et al.* [37] as hours post-fertilization.

## 2.9. Microinjection

Reagents were diluted with Danieau's buffer and phenol red to desired concentrations. All reagents were injected at the 1-cell stage unless otherwise stated. Embryos were immobilized on a 1% agar plate, and an injection pipette was forced into the area adjacent to the blastomeres. A solution of the desired volume was injected using a Nanoliter injector (World Precision Instrument, Sarasota, FL, USA).

## 2.10. Rhodamine-phalloidin staining and confocal imaging analysis

Fixed embryos were washed with PBS, blocked with 10% FBS, treated with 0.3% Triton X-100 in PBS and then incubated with rhodamine-phalloidin (Invitrogen) overnight at 4°C. After washing with PBS, nuclei were labelled with 4′,6-diamidino-2-phenylindole (DAPI, Sigma) and mounted. Fluorescence signals were recorded with the Zeiss LSM780 Confocal Microscope Imaging System (Carl Zeiss, Jena, Germany). The z-stack analysed a total of 132 μm thickness with a 1.5 μm interval.

## 2.11. Whole-mount *in situ* hybridization

Embryos were fixed with fresh 4% paraformaldehyde (PFA) overnight and manually dechorionated. WISH was performed using DIG-labelled, SP6-RNA polymerase-made riboprobes according to a previous study [38]. Stained embryos were mounted in cellulose and observed under a stereomicroscope system (Mz75, Leica Microsystems). Photographs were taken using a digital camera (Coolpix 995, Nikon, Melville, NY, USA).

## 2.12. Measurement of degrees in convergence and extension

Convergent extension during gastrulation was assessed by measuring changes in the angles formed by respective expression patterns of marker genes [39,40]. Briefly, WISH against *cathepsin L 1b* (*ctsl1b*) and *distal-less homeobox 3* (*dlx3*) were used to label prechordal plate and ectoderm borders, respectively, which formed a V-shaped pattern. We measured the angle of the V-shape to estimate the level of convergence. We also used WISH against *ctsl1b* and *no tail* (*ntl*) to mark the prechordal plate and presumptive notochord, respectively, to evaluate the dorsal axis extension in the bud stage. We enclosed each embryo within a rectangle, with the crossing of two diagonals being set to the exact middle of an embryo. A line was drawn from the anterior front of the prechordal plate to the centre of the embryo and then its centre was connected to the posterior ends of the tail bud. The angle towards the dorsal side was measured to estimate the length of the dorsal body axis, which was an indication of dorsal extension.

## 2.13. Time-lapse DIC imaging and analysis

Dechorionated embryos were mounted with 1% low-melting-point agar, and cell migration in the prechordal site and lateral site were recorded by the DIC system with the Leica DM5000B system under a 63× water immersion objective during gastrulation. The migration process was recorded at 10 s intervals for 10 min and analysed with Simple PCI Imagine System software (Compix, Sewickley, PA, USA) to track migration characteristics (migrating rate, velocity, cell protrusion numbers, tortuosity and direction of cell migration).

## 2.14. RacI and Cdc42 activation assay

The levels of activated RacI and Cdc42 were determined using the RacI and Cdc42 G-LISA Activation Assay kit (Cytoskeleton, Denver, CO, USA) according to the manufacturer's instructions. Embryos were cultured to the bud stage and dechorionated by proteinase. In total, 60 embryos were collected in a 1.5 ml pre-chilled tube and lysis buffer added. Embryos were homogenized using a 24-gauge needle, briefly

centrifuged at $10\,000g$ for 1 min, and then immediately immersed in liquid nitrogen. Protein lysates were applied to G-LISA plates using a RacI and Cdc42-specific antibody for detecting the captured active small G-protein according to the manufacturer's directions to determine the absorbance at 490 mm.

## 2.15. Generation of CRISPR/Cas9-mediated fak1a knockout fish

Three *fak1a* guiding (g)RNAs (see sequences in electronic supplementary material, table S2) were designed using the software on the CHOPCHOP website (https://chopchop.rc.fas.harvard.edu/). The gRNA template was synthesized according to a previous study [41], including annealing with two oligos and filled-in single-stranded DNA overhangs with T4 DNA polymerase (New England Biolabs, Ipswich, MA, USA). Double-stranded DNA was used as a template to synthesize gRNA by *in vitro* transcription using the MEGAscript T7 Transcription Kit (Ambion, Naugatuck, CT, USA). A zebrafish codon-optimized Cas9 pCS2+ plasmid (a kind gift from Dr Alex Schier, Harvard University) was linearized and *in vitro*-transcribed with the mMESSAGE mMACHINE SP6 kit. Cas9 mRNA (150 pg) and gRNA (50 pg) were co-injected into 1-cell-stage embryos and raised to adulthood as the F0 generation. F0 fish were crossed with wild-type (WT) fish to generate heterozygous F1 fish. F1 fish were genotyped. Briefly, tail fins were clipped and incubated in lysis buffer (0.4 mg ml$^{-1}$ proteinase K in TE buffer) at 55°C for 2 h. Samples were inactivated at 85°C for 15 min and then purified to serve as PCR templates. DNA fragments with potential mutations were amplified by a PCR using the primer set, Fak1a E10 F/R (see sequences at electronic supplementary material, table S1), cloned by TA cloning and subjected to Sanger sequencing to identify potential mutant alleles. Heterozygous F1 fish with identical alleles were incrossed to generate F2 fish. At least four batches of offspring embryos from different founders were then raised to adulthood and further analysed by a PCR using the Fak1a E10 F/R primer pairs following by Hae III digestion. Stable F2 fish were maintained for breeding. F3 homozygous embryos were used for phenotypic observations and molecular analysis.

## 2.16. Hae III mutagenesis assay and sequencing

The genomic DNA from adult tail fins or embryos was extracted by TE buffer with 0.2 mg ml$^{-1}$ proteinase K, incubated at 55°C for 2 h, and inactivated at 85°C for 10 min. A 513 bp DNA fragment was amplified by a PCR using primers as shown in electronic supplementary material, table S1. PCR products were cleaned using a PCR clean-up kit (Qiagen, Hilden, Germany) and digested with Hae III (New England Biolabs, Ipswich, MA, USA) for 2 h at 37°C. Digested products were examined by electrophoresis in a 1% agarose gel.

## 2.17. Statistical analysis

All datasets are presented as the mean ± s.e.m. Results were further analysed by a one-way analysis of variance and post-tested using Tukey's multiple-comparison tests or Student's *t*-test.

# 3. Results

## 3.1. Zebrafish Fak1a is a functional conserved homologue of human FAK

FAK is a critical factor for cell proliferation, motility and migration [42–45]. To determine whether zebrafish Fak has conserved properties, we subcloned zebrafish *fak1a* and its dominant-negative form, *frnk1a* (see the FAK domain structure in electronic supplementary material, figure S3) into a human influenza hemagglutinin (HA)-tagged pKH3 or a pEGFP-C3 vector. We successfully overexpressed them independently in 293T cells as shown by western blotting against HA or GFP (figure 1a) and found that *fak1a* and *frnk1a* were co-localized with paxillin in focal contacts (figure 1b). Cell proliferation was significantly enhanced in *fak1a*-overexpressing cells, but not in *frnk1a*-overexpressing or control *pEGFP*-transfected cells (figure 1c; $p < 0.001$). Similarly, cell transmigration activity was higher in *fak1a*-overexpressing cells (figure 1d; $p < 0.001$). These results suggest that zebrafish Fak1a is functionally conserved compared to other FAK orthologues.

To examine whether key phosphorylation sites are conserved in zebrafish Fak1a, the HA-tagged *fak1a* and *frnk1a* were overexpressed in $FAK^{-/-}$ cells, immunoprecipitated and immunoblotted. As expected, we found that Fak1a, but not Frnk1a, was autophosphorylated at Y397 (electronic supplementary material, figure S4A). The phosphorylation of Fak1a, but not Frnk1a, was notably enhanced at Y397, Y576/577, S732 and Y861 as it was in chicken orthologues, thereby demonstrating that these sites are biochemically conserved on Fak1a (electronic supplementary material, figure S4B).

## 3.2. *Fak1a* and *fak1b* are maternally and ubiquitously expressed in zebrafish embryos

To determine the *bona fide* expression pattern of *fak1a* during zebrafish embryogenesis, *fak1a* mRNA was detected by whole-mount *in situ* hybridization (WISH) analysis. Results showed that *fak1a* mRNA was maternally expressed during the 1-cell stage and ubiquitously expressed through blastulation and gastrulation (electronic supplementary material, figure S5A). In addition, *fak1a* mRNA and protein were detected by RT–PCR (electronic supplementary material, figure S5B) or immunoblotting (electronic supplementary material, figure S5C), respectively. We observed that *fak1a* mRNA and protein were constantly expressed in embryos from 30% epiboly to 17-somite stages. Furthermore, we also examined the expression patterns of another *fak* isoform, *fak1b*. *fak1b* was constantly expressed from 1-cell stage to 5 days post-fertilization by RT–PCR analysis (electronic supplementary material, figure S6A). Spatially, like *fak1a*, it was ubiquitously expressed during early embryogenesis and localized to the anterior head region at later development by WISH analysis (electronic supplementary material, figure S6B).

## 3.3. Loss of Fak1a causes gastrulation defects

To assess the developmental role of Fak1a, we microinjected 1-cell-stage zebrafish embryos with or without an antisense translation blocking MO (tMO$_1$) targeting *fak1a*. The Fak1a protein was significantly reduced to approximately 30% in

royalsocietypublishing.org/journal/rsob  Open Biol. **10**: 190273

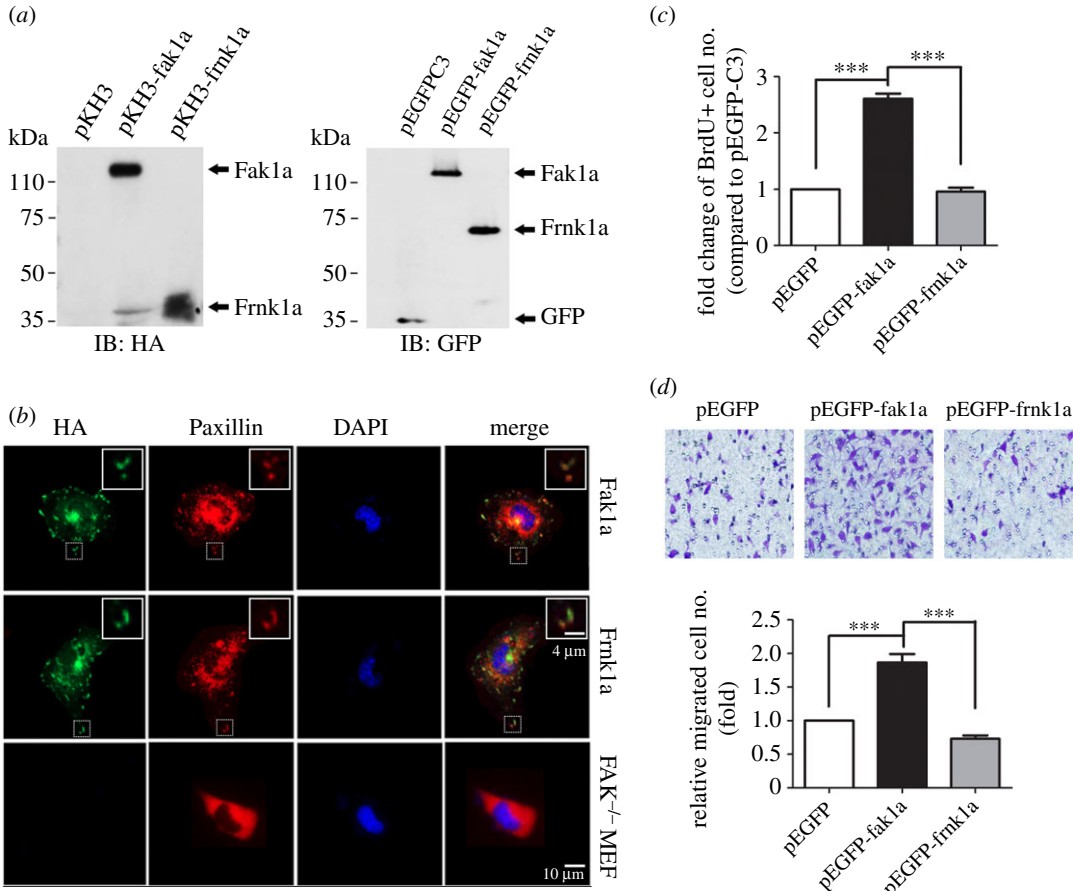

**Figure 1.** Zebrafish Fak1a is functionally conserved with mammalian FAKs. (*a*) Zebrafish Fak1a and Frnk1a were successfully expressed in 293T cells as characterized by immunoblotting (IB) using anti-HA (left) and anti-green fluorescent protein (GFP) antibodies (right). (*b*) Zebrafish Fak1a and Frnk1a with an HA tag were expressed in $FAK^{-/-}$ MEF cells as revealed by HA immunostaining shown in green (left column). They were present in focal contacts (boxed), and an enlarged image is shown at the top right corner of the respective panel. Focal contacts and cell nuclei were visualized by Paxillin (second column) and DAPI staining (third column), respectively. Merged images for Fak, Paxillin and DAPI staining are presented in the right column with enlarged insets for the boxed regions to better show the co-localization of Fak and paxillin in focal contacts. Cells were transfected with pEGFP-C3 vector only, *fak1a* or *frnk1a* to examine cell proliferation (*c*) and cell migration (*d*) as determined by a BrdU incorporation assay and Bodyen chamber migration assay, respectively (*n* = 3; ***$p < 0.001$). Detailed protocols are described in 'Material and methods'.

*fak1a* tMO₁-injected embryos compared to untreated ones (figure 2*b*). MO-injected embryos are called morphants hereafter. In *fak1a* morphants, embryos displayed a cylinder-like shape compared to the normal ball-like form. The progression of epiboly in these morphants was notably slower during gastrulation. The progression of epiboly was either delayed and eventually reached 100% epiboly (epiboly delay), or arrested. The movements of the deep cell layer (DCL) and enveloping layer (EVL) were asynchronous (figure 2*c*). Overall, we found 47% and 18% of *fak1a* tMO₁ morphants had delayed and arrested epiboly, respectively (figure 2*d*). Among the arrested embryos, 87% of them did not reach 100% epiboly at 16 hpf. This was not due to the toxicity of the MO, since no epiboly defect was observed in embryos injected with 10 ng of the N-25 control MO (electronic supplementary material, figure S7). To further test the specificity of the phenotype, *fak1a* mRNA lacking the target site of the *fak1a* tMO₁ was prepared and injected with or without *fak1a* tMO₁. Relatively lower percentages (approx. 30%) epiboly defective embryos were observed in embryos injected with 125 pg *fak1a* mRNA, but it significantly rescued the epiboly defect when co-injected with the *fak1a* tMO₁ (figure 2*c*, left). The injection of *fak1a* mRNA at a higher dose (250 pg) further reduced the epiboly defects observed in *fak1a* morphants (electronic supplementary

material, figure S7). The rescued embryos appeared mostly normal compared to control embryos (electronic supplementary material, figure S8). In addition, we also found dose-dependent epiboly defects in embryos injected with dominant-negative *frnk1a* mRNA (figure 2*c*, right). Furthermore, we investigated how the loss of *fak1a* affecting its downstream effectors and found that the levels of phosphorylation of paxillin, p38 mitogen-activated protein kinase (p38) and Akt were all reduced in *fak1a* morphants (figure 2*d*).

To ensure the MO-induced epiboly defects are specific to the loss of Fak1a, we designed and tested a second translational blocking *fak1a* MO (tMO₂), which does not overlap the tMO₁ target site, and found the *fak1a* tMO₂ also caused 70% delayed and 5% arrested epiboly (electronic supplementary material, figure S9A, B). Effects of the *fak1a* tMO₂ were also dose-dependent and could be rescued by the co-injection of *fak1a* mRNA (electronic supplementary material, figure S9C). Lastly, the standard control MO (figure 2*c*) and a random control MO (N-25) (data not shown) were both tested at up to 10 ng per embryo with no notable effect on epiboly or gastrulation. Therefore, untreated embryos were used as controls hereafter unless otherwise specified. Furthermore, the *fak1a* tMO₁ was relatively more effective, thus we used it for the rest of the experiments.

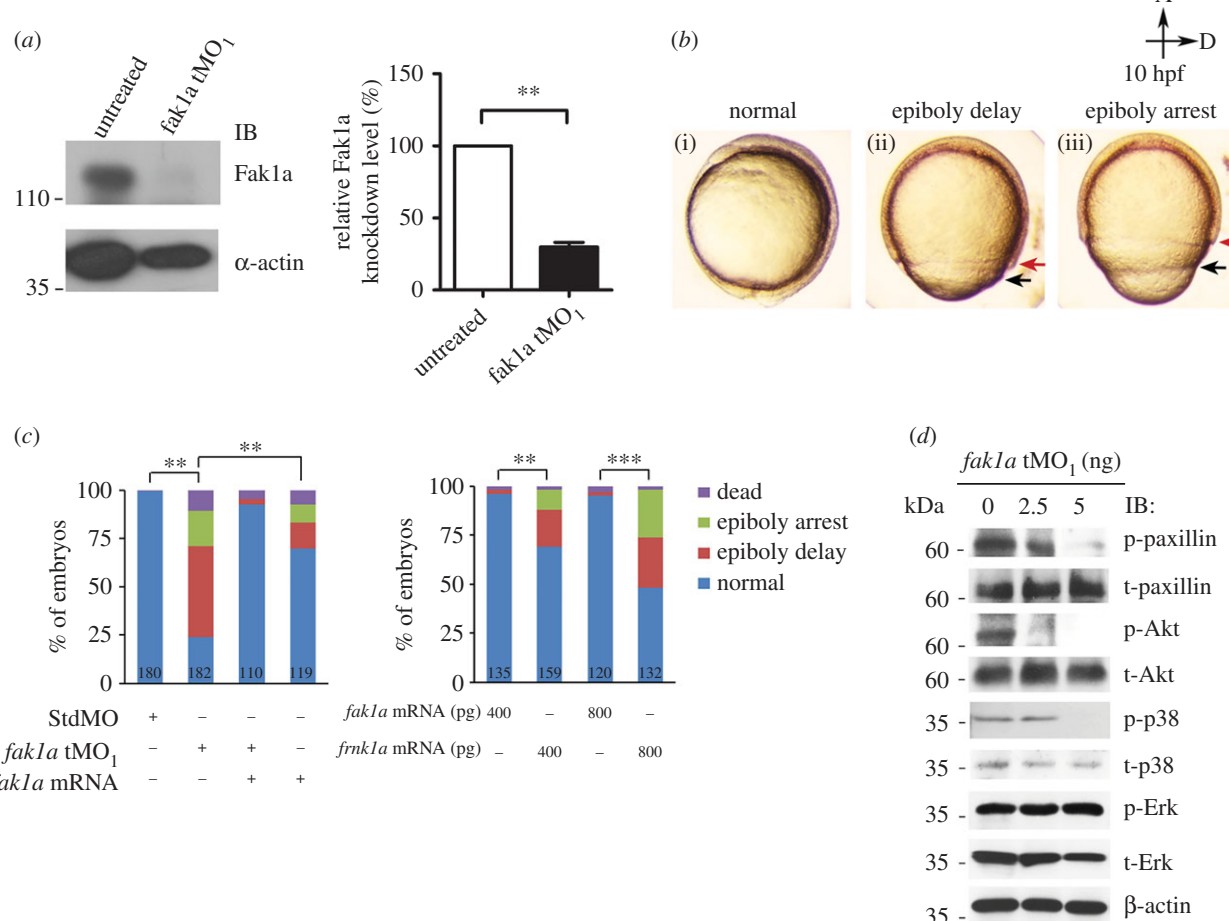

**Figure 2.** Zebrafish *fak1a* morphants reveal severe gastrulation defects. (*a*) Zebrafish embryos were treated or untreated with a *fak1a* translation blocking morpholino (*fak1a* tMO$_1$) and immunoblotted (IB) against Fak1a and β-actin (loading control). The intensities of Fak1a bands were normalized to that of β-actin. The relative knockdown levels of Fak1a in treated embryos are shown by comparing band intensity to that of untreated embryos (right panel, $n = 3$; **$p < 0.01$). (*b*) The epiboly progression of *fak1a* tMO$_1$-treated embryos was either delayed (ii) or arrested (iii) compared to the normal standard control MO-injected embryos (i) at 10 hpf as presented in bright-field images. Black and red arrows point to the running fronts of the enveloping layer and deep cell layer, respectively. All images have the animal pole (A) placed at the top and the dorsal to the right (D). (*c*) Embryos were treated as indicated and classified into different categories as indicated, and percentages of embryos in each category are shown ($n = 3$, **$p < 0.01$; ***$p < 0.001$). The total number of embryos used in each treatment is shown at the bottom of each bar. (*d*) Proteins of embryos injected with or without 2.5/5 ng *fak1a* tMO$_1$ were extracted and immunoblotted using indicated antibodies. β-Actin was used as a loading control.

## 3.4. Loss of Fak1a causes reduction in cortical actin fibres and uneven distribution of YSL nuclei

Filamentous (F)-actin forms a ring-like structure at the leading edge of the EVL, where it links the EVL and the yolk cell to provide the pulling force for blastoderm during epiboly [46] and is pivotal for gastrulation [36,47]. Thus, we examined the organization of F-actin of 75%-epiboly-stage embryos by probing with rhodamine-phalloidin. In addition, we also counterstained cell nuclei by 4′,6-diamidino-2-phenylindole (DAPI) and Fak1a by immunostaining. Immunostaining showed the ubiquitous presence of Fak1a in blastodisc and YSL in control embryos, but the Fak1a staining was dramatically reduced in *fak1a* tMO$_1$ morphants (figure 3*a*). This further confirmed the efficacy of knockdown effect by the *fak1a* tMO$_1$. The staining also clearly marked the EVL and DCL as indicated by arrows. In control embryos, the EVL and the DCL migrated coherently with a 92.8 ± 14.3 μm gap in between. By contrast, the gap was significantly increased to 243.6 ± 33.3 μm in *fak1a* tMO$_1$ morphants (figure 3*b*). Only about 20% of embryos with a gap smaller than 100 μm that indicates epiboly was significantly disturbed in most tMO$_1$ morphants (figure 3*c*, $p <$

0.001). Furthermore, by examining the region flanking the margin of the EVL at a higher magnification, we found that F-actin fibres were not tightly distributed in *fak1a* tMO$_1$ morphants compared to control embryos. Rough and uneven blastoderm margins were observed in *fak1a* morphants. This suggests that the loose actin network may lead to un-synchronized movements of the EVL and DCL in *fak1a* tMO$_1$ morphants.

To reveal the formation of the junction between the blastoderm and actin cap, we further examined the structure of F-actin in 60%-epiboly-stage embryos. YSL nuclei were evenly scattered at the EVL margin in untreated embryos, but unevenly distributed in *fak1a* morphants (figure 3*d*, blue nuclei labelled by yellow asterisks). Uneven blastoderm margin and distribution of F-actin were also found in *fak1a* tMO$_1$ morphants. We also observed previously reported actin-based structures, called actin bundles, in the vegetal cortex of the yolk cell that connect the vegetal cortex with the external yolk syncytial nuclei (eYSN) in a vegetal-to-animal direction (indicated by asterisks in figure 3*d*; Li *et al.* [48]). A significant loss of actin bundles along with a loose connection of the actin ring to the cap was seen in *fak1a*

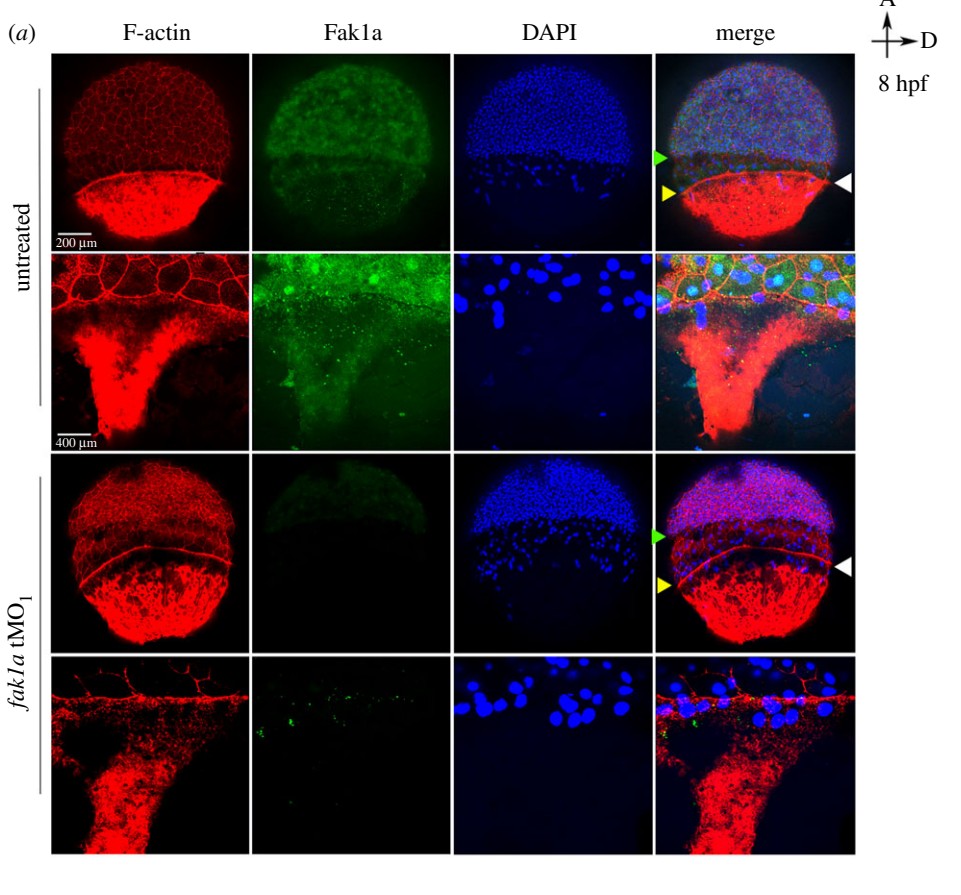

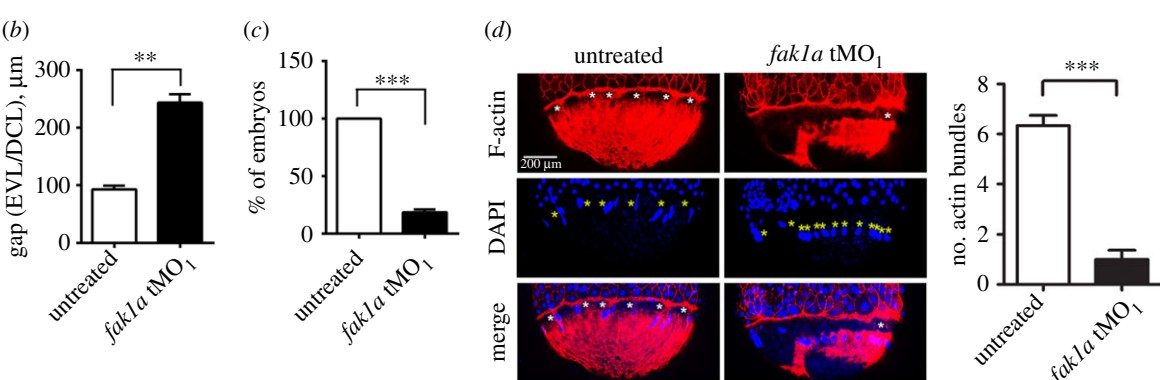

**Figure 3.** Loss of Fak1a perturbs the synchronized migration of enveloping and deep cell layers and the F-actin network. (a) Embryos untreated or injected with 5 ng of a *fak1a* translation blocking morpholino (tMO$_1$) were fixed at 8 hpf and subjected to F-actin/DAPI staining and Fak1a immunohistochemistry. Embryos were examined and photographed for the whole embryo image (upper) or a region flanking the yolk syncytial at a higher magnification (lower) under confocal microscopy. Representative photographs of different channels and merged images are shown. White arrowheads point to actin rings. Yellow and green arrowheads point to the running fronts of the enveloping (EVL) and deep cell layers (DCL), respectively. (b) Graphic demonstration of the average gap between EVL and DCL in untreated and fak1a tMO1-treated embryos ($n = 3$; $n = 30$, **$p < 0.01$). (c) The EVL/DCL gap smaller than 100 μm were considered normal, and the percentages of normal embryos are presented ($n = 3$; $n = 30$, ***$p < 0.001$). (d) The actin bundles (marked by white asterisks) between the actin ring and vegetal actin cap were clearly reduced in *fak1a* morphants, and the numbers of actin bundles are quantified in the right panel ($n = 3$; $n = 10$, ***$p < 0.001$). Disorganized YSL nuclei (yellow asterisks) were also observed in *fak1a* morphants (see DAPI staining and merged images). As indicated by the arrows in the top right corner, all images have the animal pole (a) placed at the top and the dorsal to the right (d).

morphants. These data indicate that Fak1a is critical to the arrangement of cortical actin during epiboly.

## 3.5. Loss of Fak1a causes convergence and extension defect

We further analysed whether the loss of Fak1a causes convergent extension defects in the bud-stage embryos by WISH analysis. *cathepsin L 1b* (*ctsl1b*) and *distal-less homeobox 3*

(*dlx3*) were used to label the prechordal plate and ectoderm borders, respectively. Staining gave rise to a V-shaped pattern as shown in figure 4a (left column), and the angle of the V-shape was measured to estimate the level of convergence in all embryos. The average angles of convergence were dose-dependently increased from 77.8° in untreated to 104° in 0–5 ng *fak1a* tMO$_1$-injected embryos (figure 4b, left panel). The dose-dependent convergence defect was also observed in tMO$_2$ morphants (electronic supplementary material, figure S9D).

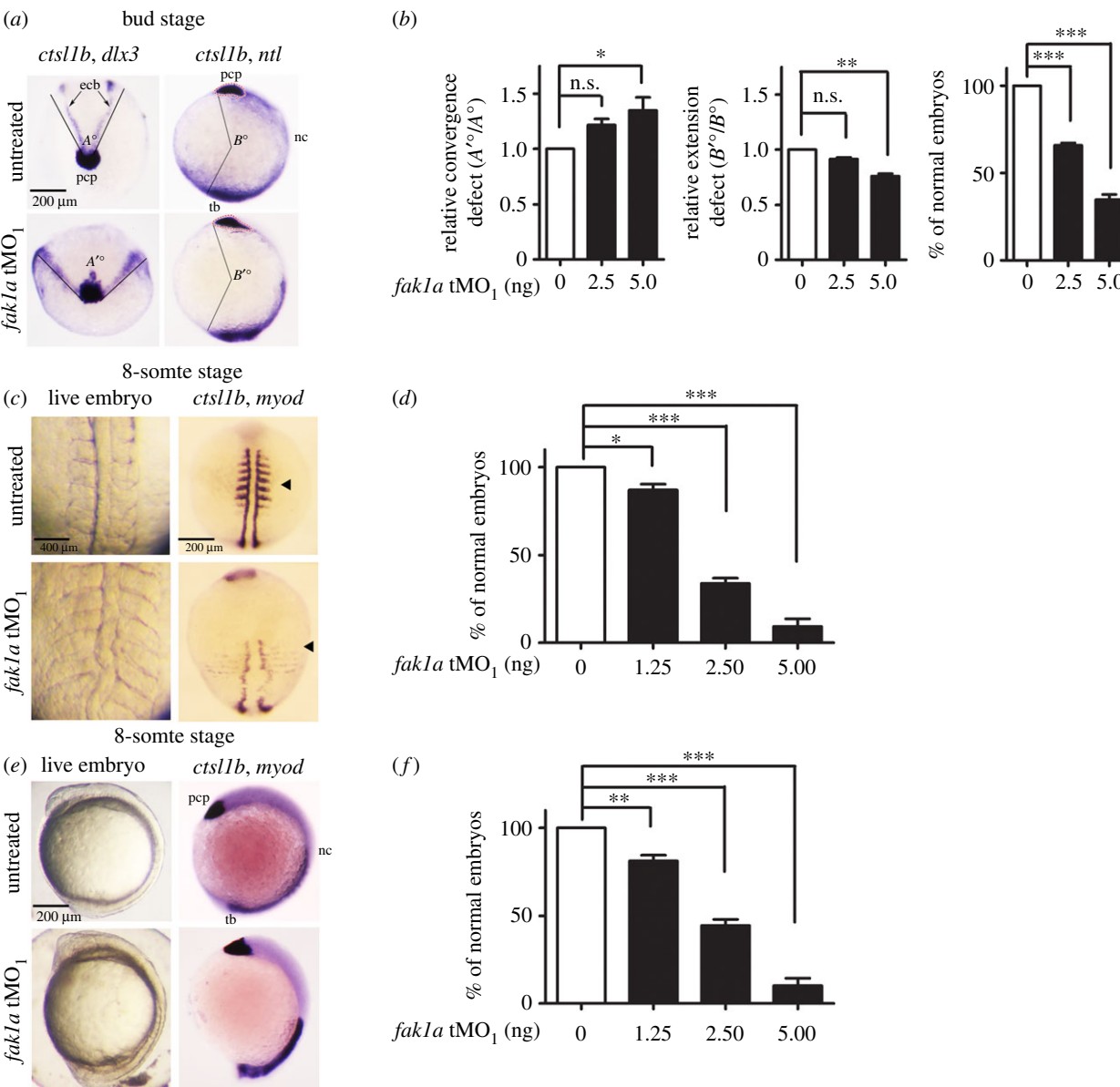

**Figure 4.** Loss of Fak1a causes *abnormal* convergence and extension movements. (*a*) Embryos injected with a designated amount of a *fak1a* translation blocking morpholino (tMO₁) were subjected to WISH against indicated genes at the bud stage. Representative WISH staining photographs against *ctsl1b/dlx3* or *ctsl1b/ntl* are presented for untreated and *fak1a* tMO₁-treated embryos (*fak1a* tMO₁). *ctsl1b*, *dix3* and *ntl* staining were used to label the prechordal plate (pcp), ectodermal borders (ecb) and notochord (nt)/tail bud (tb), respectively. For simplicity, tissues are labelled in untreated embryos only. To reveal dorsal convergence, embryos were probed with *ctsl1b/dlx3* as shown in the anterior view (left). A round prechordal plate is in the middle with two ectodermal borders forming a V-shape. Lines were drawn along the ectodermal borders that matched the prechordal plates. The V-shape formed an angle as indicated by $A°$ and $A'°$ in untreated and *fak1a* tMO₁-treated embryos, respectively. To reveal the dorsal extension, embryos were probed with *ctsl1b/ntl* as shown in the lateral view with the dorsal to the right (right). Lines were drawn from the anterior front of the prechordal plate and the posterior end of the tail bud to the centre of the embryo forming an angle towards the dorsal as indicated by $B°$ and $B'°$ in untreated and *fak1a* tMO₁-treated embryos, respectively. (*b*) The relative convergence and extension defects were quantified by calculating the ratios of $A'°/A°$ (left panel) and $B'°/B°$ (middle panel), respectively. The normality of anterolateral migration of the prechordal plate was examined by signals of *ctsl1b* and *dlx3* in embryos injected with a designated amount of the *fak1a* tMO₁ (right panel). $n = 3$, *$p < 0.05$, **$p < 0.01$, ***$p < 0.001$, n.s., not significant. (*c*) Embryos were injected with different amounts of the *fak1a* tMO₁, cultured to the 8-somite stage, photographed (live embryo) or fixed, and subjected to WISH against *ctsl1b/myo D*. Representative dorsal view photographs are shown in (*c*), and statistical comparisons of embryos with normal somites are presented in (*d*). As described in (*c,d*), the anterior and posterior extensions of the dorsal axes were examined. Representative lateral-view photographs with the anterior to the left are shown in (*e*), and statistical comparisons of embryos with normal dorsal axis extension are presented in (*f*).

To evaluate the dorsal axis extension, we used WISH against *ctsl1b* and *no tail* (*ntl*) to mark the prechordal plate and presumptive notochord, respectively. We drew a line from the anterior front of the prechordal plate to the centre of an embryo (see the determination of centre in Material and Methods), and then connected it to the posterior end of the tail bud (figure 4*a*, right column). The angle towards the dorsal was used to estimate the length of the dorsal body axis, which reflects the degree of dorsal extension. The average angles of extension were 239.2°–182.4° in 0–5 ng *fak1a* tMO₁-injected embryos. *fak1a* tMO₁ morphants showed significantly reduced angles compared to those in control embryos, implying an inhibition of extension (figure 4*b*, middle graph; $p < 0.01$). A similar defect in extension was also observed in tMO₂ morphants (electronic supplementary material, figure S9D).

In control embryos, individual *ctsl1b*-expressing cells consolidated and coordinately moved anterior to the frontal edge of the neural plate (expressing *dlx3*) as the tissue advanced

beyond the animal pole. In *fak1a* morphants, the *ctsl1b*-expressing cells failed to form a consolidated group and abnormally dispersed anteriorly–posteriorly (figure 4*a*, right column). Mesendodermal cells did not acquire the coordinated crescent shape anterior to the neural plate by the tail bud stage and exhibited a more diffused *dlx3* expression. These phenomena illustrate that Fak1a is required for anterior movement and consolidation of the polster along with the simultaneous convergence and extension of posterior tissues.

We also observed an expanded somite width and a shortened body axis in *fak1a* morphants at the 6-somite stage. The tail rudiment of *fak1a* morphants was either less extended or stalled at the vegetal pole (figure 4*c,e*, left). These phenotypes are typical defects of convergence and extension, so we fixed both untreated embryos and *fak1a* morphants for WISH against *myogenic differentiation 1* (*myod*) and *ctsl1b* to reveal somites and the prechordal plate, respectively (figure 4*c,e*, right column). The percentage of embryos with a normal somite width and body axis dose-dependently decreased in *fak1a* morphants (figure 4*d,f*). This further confirms the deteriorating effect on the convergent extension in the absence of *fak1a*.

## 3.6. No synergy exists between fak1a and fak1b in regulating gastrulation

To examine the role of another zebrafish Fak, Fak1b, during gastrulation, we applied two MOs, *fak1b* tMO$_1$ and *fak1b* tMO$_2$ in zebrafish embryos. In general, both *fak1b* MOs caused dose-dependent epiboly delay and arrest. The inhibition of *fak1b* tMO$_2$ was much weaker compared to that of *fak1b* tMO$_1$ (electronic supplementary material, figure S10A, B). In addition, both *fak1b* MOs resulted in significant convergent extension defects (electronic supplementary material, figure S10C,D).

To investigate the synergy of Fak1a and Fak1b during gastrulation, we co-injected *fak1a* tMO$_1$ and *fak1b* tMO$_1$ at two different subthreshold dosages into embryos. The injections of both MOs at 1.25 ng and 2.5 ng only showed additive defects during gastrulation (electronic supplementary material, figure S11). These results suggested no synergy existed between *fak1a* and *fak1b* in regulating gastrulation.

## 3.7. Loss of Fak1a impedes the migration of hypoblast cells during gastrulation

*fak1a* MO-induced gastrulation defects were plausibly caused by disturbance of cell migration. To examine cell migration *in vivo*, we performed a time-lapse mobility assay on the running fronts of involuting prechordal plate cells (electronic supplementary material, movies S1 and S2) and convergent lateral cells (electronic supplementary material, movies S3 and S4) during the 75–90% epiboly stage. Cell migration of prechordal plate cells and lateral cells were both inhibited in *fak1a* morphants (*fak1a* tMO$_1$) compared to those in random control MO N-25-injected embryos. We recorded and quantified the migration rate, migration velocity, protrusion number and direction of tracked cells using Simple PCI software (figure 5*a*). To plot the trace of each migrating cell, the origin (o′) was defined as the location of a cell at the beginning of the recording. The anterior of the embryo (a′) or animal pole (a) is indicated by an arrow on the *y*-axis in each plot. The arrow of the *x*-axis is perpendicular to the

anterior axis in the prechordal cell tracking plot or points to the dorsal side in the lateral cell tracking plot. Each arrow represents the migration route of one cell. The arrow points towards the direction of movement and the length of the arrow is the distance of cell migration (figure 5*b*). The direction of cell migration was determined by measuring the angle between the *y*-axis and each migrating trace (figure 5*b*). Quantitative data are shown in figure 5*c*. Cells migrated more unidirectionally at the mean converged angles of 14.7° and 44.7° in prechordal cells and lateral cells, respectively. By contrast, *fak1a* morphant cells displayed uncoordinated movements at the mean converged angles of 26.7° and 82.7° in prechordal cells and lateral cells, respectively. The migration velocity (the distance between the origin and the end of the trace (DOE) divided by time) of prechordal cells of *fak1a* morphants (0.28 µm s$^{-1}$) was significantly slower than in control embryos (0.44 µm s$^{-1}$). Similar results were also observed in the migration of lateral cells (*fak1a* morphants: 0.37 µm s$^{-1}$, control embryos: 0.51 µm s$^{-1}$; figure 5*d*, left; $p < 0.01$). A significant difference was also shown in the prechordal cell migration rate (length of the migrating trace (LMT) divided by time) in *fak1a* morphants (0.1 µm s$^{-1}$) and controls (0.44 µm s$^{-1}$; $p < 0.001$), but no effect was seen on lateral cells (figure 5*d*, middle left). Tortuosity (LMT/DOE) was not affected in prechordal cells but was reduced in lateral cells of *fak1a* morphants (morphants: 2.37, control embryos: 3.6) (figure 5*d*, middle right). These observations clearly demonstrate the inhibition of cell migration during gastrulation by *fak1a* tMO$_1$.

We further compared cell protrusions between control embryos and *fak1a* morphants and found that the average numbers of newly generated protrusions of *fak1a* morphants were significantly lower than those of control embryos in both cell groups (figure 5*d*, right; prechordal cells: morphant (6.26), control (8.67), $p < 0.05$; lateral cells: morphant (5.7), control (8.89), $p < 0.01$). These data suggest that Fak1a is required for proper cell protrusions to drive effective cell movements during gastrulation.

## 3.8. FAK1a autonomously mediates cell protrusions, but non-cell autonomously regulates the directions of cell movement during gastrulation

To assess cell-autonomy of Fak1a function in mediating migration of lateral hypoblast cells, we performed cell transplantation experiments by injecting rhodamine dextran into donor embryos as a cell tracer. Rhodamine-labelled fluorescent blastomeres from StdMO or *fak1a* morphants were transplanted to non-fluorescent host embryos treated without or with *fak1a* tMO$_1$, examined under epifluorescence microscopy and recorded for 30 min. The extensions of cell body were designated as protrusions and the direction and numbers of these protrusions were measured at each cells. Under microscopy, cells from a StdMO morphant showed highly protrusive activities (4.1 ± 0.9 protrusions, *n* = 39) with well-formed protrusions when transplanted into an untreated host embryo ('STD > UT', arrow, figure 6*a*; electronic supplementary material, movie S5). Cell protrusions remained active in those cells transplanted to a *fak1a* morphant host ('STD > MO', arrows, figure 6*b*; electronic supplementary material, movie S6). Moreover, protrusions were active in cells from a *fak1a* morphant transplanted into

royalsocietypublishing.org/journal/rsob Open Biol. 10: 190273

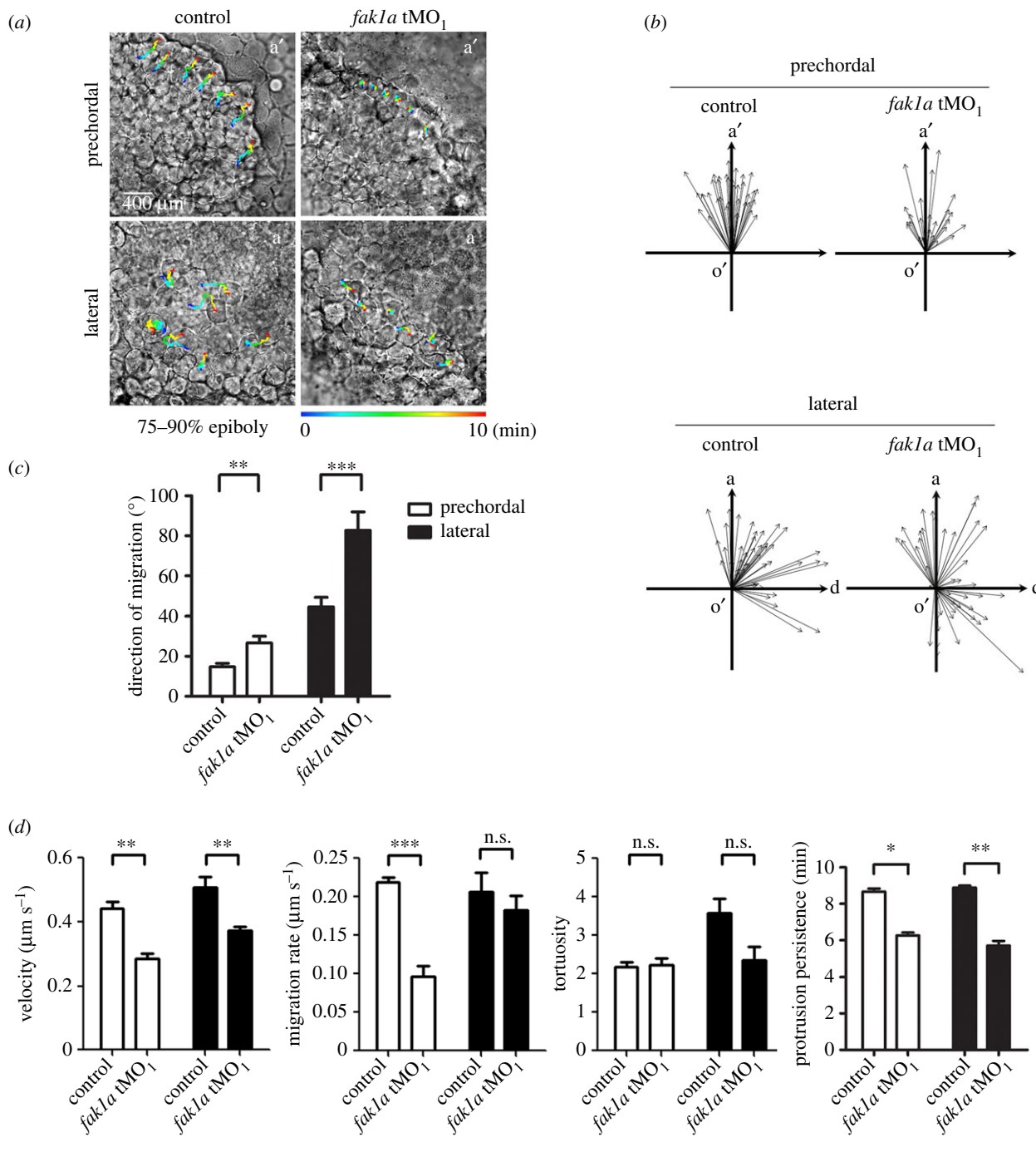

**Figure 5.** Loss of Fak1a perturbs hypoblast cell migration. Embryos were injected with 5 ng of a random control morpholino (MO) N-25 (control) or *fak1a* translation blocking MO (tMO₁), immobilized and monitored under differential interference microscopy. Time-lapse movies were taken for 10 min during the 75–90% epiboly stage to reveal the involuting cell migration of the anterior prechordal plate or convergent movement of lateral cells (see electronic supplementary material, movies S1–S4). (*a*) Representative snapshots of the prechordal plate or lateral cells at the end of representative recordings are shown. More than six cells were selected from an embryo to be traced in each movie, and their migrating routes are depicted by a rainbow line representing the recording time at 0–10 min. a′, anterior; a, animal pole. These experiments were repeated at least three times. (*b*) The moving direction (arrow direction) and migration distance (arrow length) of each traced cell are represented by an arrow. The origin (o′) of the coordinate plane stands for the starting point of each cell. a′, anterior; a, animal pole; d, dorsal side. (*c*) The polarity of each cell was measured as described in Results, and the analysis showed a significant loss in the polarity of the prechordal and lateral cell migration in *fak1a* tMO₁-treated embryos ($n \geq 3$; $n \geq 36$). (*d*) The migration velocity, migration rate, tortuosity (route/distance) and protrusion persistence of each recording were analysed with Simple PCI software, and comparisons between groups are shown. n.s., not significant, $*p < 0.05$; $**p < 0.01$, $***p < 0.001$.

an untreated host ('MO > UT', figure 6*c*; electronic supplementary material, movie S7) compared to a reduced number of protrusions in a *fak1a* morphant host ('MO > MO', figure 6*d*; electronic supplementary material, movie S8).

To quantify the protrusion activity, we examined how many protrusions were formed in 30 min in five randomly selected cells for each embryo examined. The STD > UT group

had $4.1 \pm 0.9$ protrusions ($n = 39$) and the STD > MO group had $4.8 \pm 0.5$ protrusions ($n = 28$); the MO > UT group had $4.5 \pm 0.0$ ($n = 22$) and the MO–MO group had $2.8 \pm 0.3$ ($n = 54$). Only the MO > MO treatment showed significant less protrusion numbers compared to the other treatments ($p < 0.05$) (figure 6*e*). The STD > UT cells travelled faster and longer than those STD > MO cells (figure 6*f*). The curvilinear velocity (Vcl,

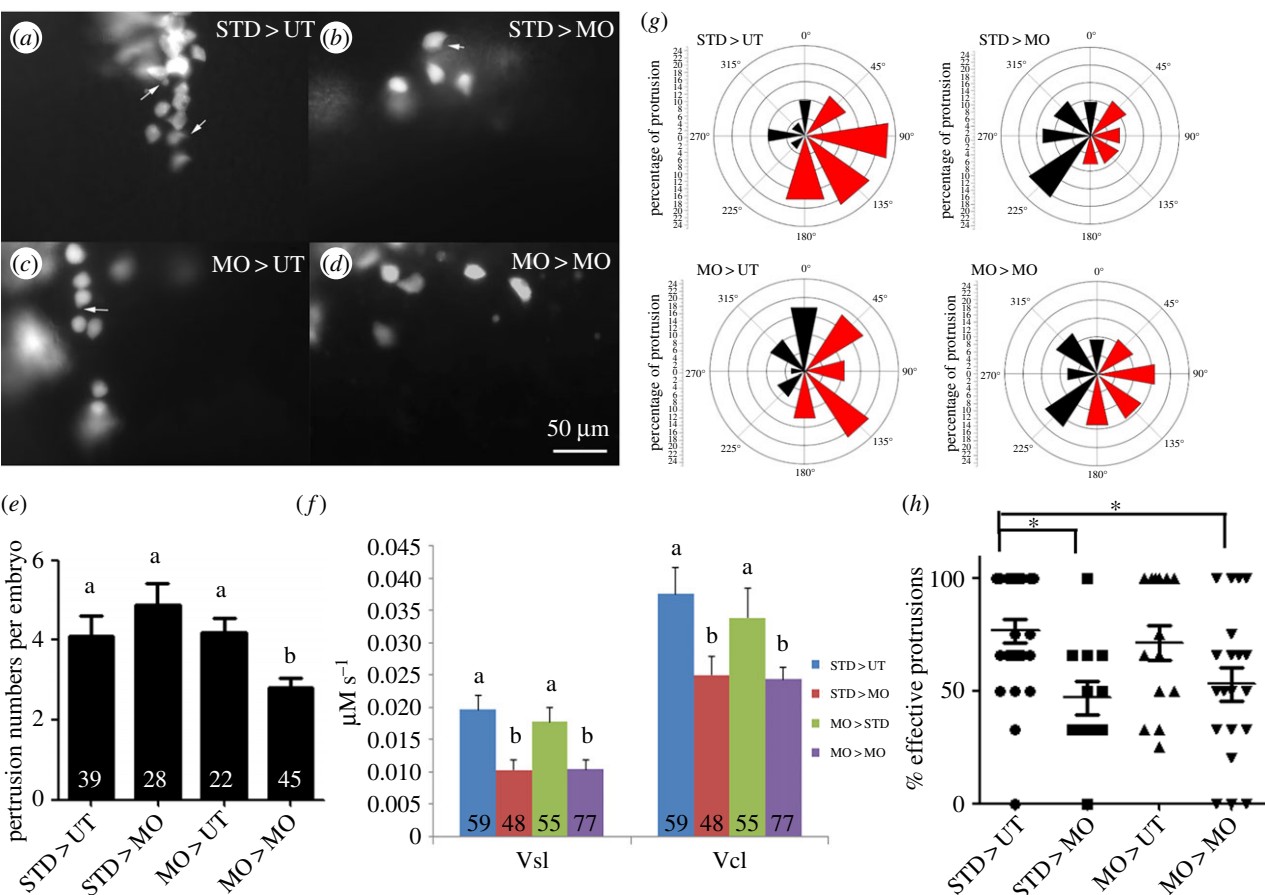

**Figure 6.** Fak1a functions non-cell-autonomously to regulate cell migration during gastrulation. (*a–d*) Rhodamine-labelled blastomeres were transplanted from embryos injected with 5 ng of StdMO (Std) or tMO$_1$ (MO) with rhodamine dextran to untreated hosts (UT) or tMO$_1$ morphant hosts (MO). Host embryos were then imaged (animal pole on the top and vegetal pole at the bottom) under epifluorescence microscopy, recorded and representative snapshots are shown in (*a*) STD > UT: StdMO-treated cells in an untreated host. (*b*) STD > MO: StdMO-treated cells in a tMO$_1$ morphant. (*c*) MO > UT: tMO$_1$-treated cells in an untreated host. (*d*) MO > MO: tMO$_1$-treated cells in a tMO$_1$ morphant. Arrows indicate the representative cellular protrusions in each embryo. (*e*) The average protrusion numbers per embryo were counted from each recording and shown. The total number of embryos used for each treatment is shown on the bottom of each bar. Values between groups with a significant difference (*p* < 0.05) are denoted by different letters. (*f*) The transplanted cells were traced and their curvillinear velocity (Vcl) and strait line velocity (Vcl) were calculated and shown. Values between groups with a significant difference (*p* < 0.05) are denoted by different letters. (*g*) The distribution of protrusions formed from donor cells transplanted to host is shown. The centre of a cell is regarded as the centroid of the rose diagram. A rose diagram is divided into eight equally parts with designated angles. The 0°, 90° and 180° points to the animal pore, dorsal side and vegetal pore, respectively. The percentages of protrusion per cell were calculated and plotted on rose diagrams. The effective protrusions were marked in red and the ineffective ones are marked in dark. *Y*-axis for the rose diagrams represents the percentage of protrusions in each direction bin. (*h*) The percentage of effective protrusions per cell in each group were shown (*n* = 3, *\*p* < 0.05).

curvilinear distance/time) of STD > UT cells was 0.038 ± 0.004 μm s$^{-1}$ (*n* = 59), but was only 0.025 ± 0.003 μm s$^{-1}$ (*n* = 48) for the STD > MO cells (*p* < 0.05). The MO > UT cells travelled with a faster speed and a longer distance in Vcl (0.034 ± 0.005 μm s$^{-1}$, *n* = 55) compared to those MO > MO cells (0.024 ± 0.002 μm s$^{-1}$, *n* = 77). Moreover, the STD > UT and MO > UT cells migrated straighter as shown by a higher straight-line velocity (Vsl, straight-line distance/time, figure 6*f*), which were 0.020 ± 0.002 μm s$^{-1}$ (*n* = 59) and 0.010 ± 0.002 μm s$^{-1}$ (*n* = 48), respectively. By contrast, STD > MO (*n* = 55) and MO > MO (*n* = 77) cells migrated less linearly with a Vsl of 0.0178 ± 0.002 μm s$^{-1}$ and 0.010 ± 0.001 μm s$^{-1}$, respectively, compared to cells in the UT hosts. It appeared that zebrafish Fak1a mediates gastrulation cell migration non-cell autonomously. However, it was puzzling that the number of protrusions formed in the STD > MO cells was not different from that of STD > UT cells (figure 6*e*). We reasoned that the directionality of protrusion might be different between groups. To determine the directionality of protrusions, the centre of a cell was considered as to be the centroid of the rose diagram. A rose diagram was divided into eight equal pies with designated angles.

The 0°, 90° and 180° referred to animal pore, dorsal side and vegetal pore, respectively. The percentage of protrusions for all cells fell into each pie was plotted on a rose diagram (figure 6*g*). Under the STD > UT condition, the majority (75%) of cells protruded towards the expected dorsal side and vegetable pole (DV, in red, between 45° and 180°), which we considered them as 'effective protrusion', with less than 25% protrusions extended towards the opposite direction (in black, between 0° and 225°), which were non-effective protrusions. The average percentages of effective protrusion were as following: STD > UT (74.6%), STD > MO (45.7%), MO > UT (61.5%) and MO > MO (55.8%) (figure 6*g*). We further analysed the percentages of effective protrusion per cell. Similarly, transplanted cells from either StdMO or *fak1a* morphants tended to have higher percentages of effective protrusions in untreated hosts than that of *fak1a* morphants hosts (figure 6*h*). These data clearly demonstrated that the directionality of protrusion was notably affected in the *fak1a* morphant but not in the untreated hosts. Thus, it suggests that zebrafish FAK1a mediates the direction of gastrulation cell migration in a non-cell-autonomous manner.

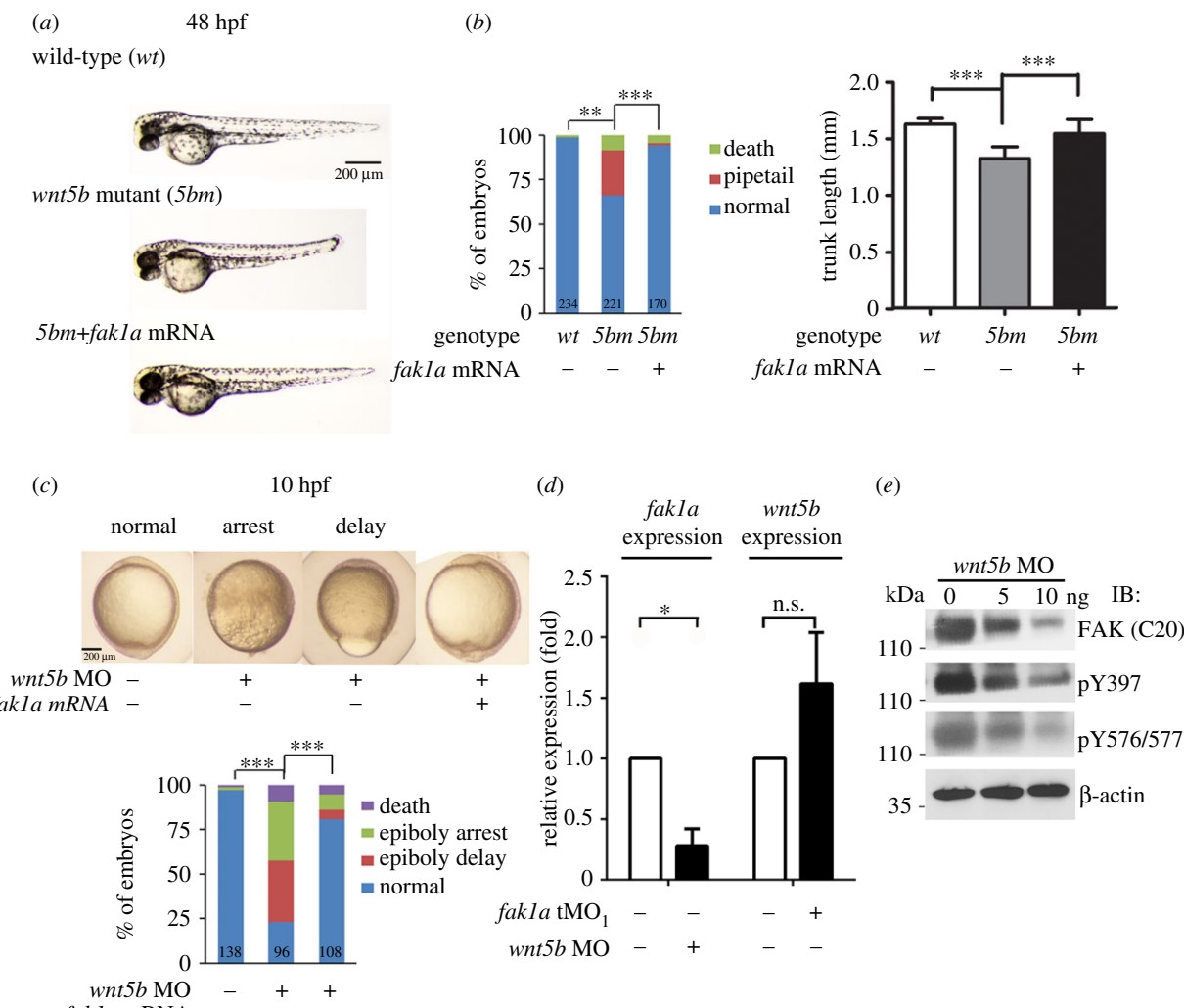

**Figure 7.** Overexpression of *fak1a* rescues gastrulation defects in *wnt5b*-deficient embryos. (*a*) Wild-type or *wnt5b/ppt* mutant embryos were untreated or injected with *fak1a* mRNA, cultured until 48 h post-fertilization (hpf), and photographed. The percentages of embryos with dead, pipetail or normal phenotypes were calculated and shown (right). The total number of embryos used for each treatment is shown at the bottom of each bar. (*b*) The average trunk lengths among the three groups were compared ($n = 3$, $**p < 0.01$, $***p < 0.001$). (*c*) Embryos were injected with 3.75 ng *wnt5b* morpholino (MO) without (control) or with 200 pg *fak1a* mRNA and photographed at 10 hpf. *wnt5b* MO-injected embryos showed different degrees of epiboly defects, which was rescued by a *fak1a* mRNA co-injection. Representative photographs are shown at the top and quantitative analysis shown at the bottom. (*d*) Embryos were injected with designated MOs, cultured to the bud stage, lysed and subjected to a qPCR to determine *fak1a* or *wnt5b* expression. (*e*) Embryos were injected with a designated amount of the *wnt5b* MO, collected at the bud stage and subjected to immunoblotting against indicated antibodies specific to focal adhesion kinase (FA) (C-20) and different FAK phosphorylation sites. β-actin served as an internal control.

## 3.9. No synergy between Wnt5b and Fak1a signalling during gastrulation

Zebrafish *wnt5b/pipetail* (*ppt*) mutants exhibit similar convergence and extension defects [29,49] as that observed in *fak1a* morphants. The knockdown of *wnt5a* expression reduced focal adhesion dynamics by affecting FAK phosphorylation in cellular assays [50]. We suspected that Fak1a may interact with the Wnt5b-mediated signalling pathway during gastrulation. Hence, we tested whether *fak1a* overexpression can rescue the convergence and extension defects of *ppt* mutants. About 25% of embryos from a standard cross of heterozygous *ppt* mutants showed convergence and extension defects with a shortened body length and a spade tail compared to normal larvae at 48 hpf (figure 7*a*). The phenotypes in *ppt mutants* injected with *fak1a* mRNA were reduced to 2% (figure 7*b*, left; $p < 0.001$). Rescued *ppt* mutants had significantly longer trunks (averaging 1.55 mm; $p < 0.001$) than uninjected siblings (1.33 mm), while WT embryos had an average trunk length of

1.63 mm (figure 7*b*, right; $p < 0.001$). Homozygous Wnt mutants are embryonic lethal and the use of heterozygous mutants only produce a quarter of homozygous mutant embryos which cannot be identified until the appearance of phenotypes that is not practical for intensive analyses. Thus, we knocked down *wnt5b* by a specific MO, which had been shown to induce gastrulation defects independent of p53 (electronic supplementary material, figure S12) [51]. We observed approximately 75% of embryos had epiboly arrest and delay phenotypes, which were not observed in *ppt* mutants and presumably due to the presence of maternal Wnt5b and the co-injection of *fak1a* mRNA significantly rescued epiboly defects in *wnt5b* morphants (figure 7*c*). The higher penetrance of *wnt5b* MO-induced phenotypes allowed us to further examine the interaction of FAK and Wnt5b during gastrulation. The co-injection of *fak1a* mRNA significantly rescued the epiboly defects in *wnt5b* morphants (figure 7*c*). To test the synergistic effect between Wnt5b and Fak1a, we co-injected subthreshold dosages of *wnt5b* and *fak1a* MOs, but found no synergistic

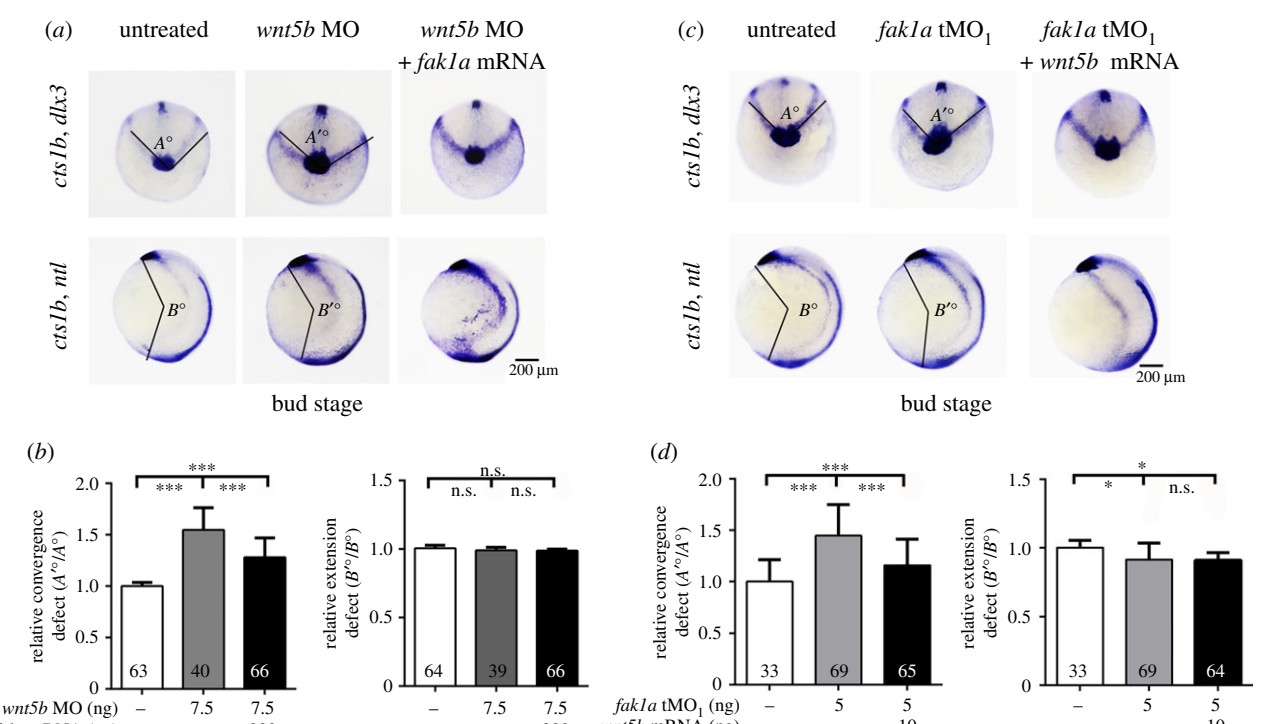

**Figure 8.** Fak1a and Wnt5b reciprocally rescue convergence defects in *wnt5b* and *fak1a* MO-injected embryos, respectively. (*a*,*b*) Embryos were untreated, injected with 7.5 ng of the *wnt5b* MO only or co-injected with both 7.5 ng of the *wnt5b* MO and 200 pg of *fak1a* mRNA. (*c*,*d*) Embryos were untreated, injected with 5 ng of the *fak1a* tMO$_1$ only, or co-injected with both 5 ng of the *fak1a* tMO$_1$ and 10 pg of *wnt5b* mRNA. Embryos were then cultured to the bud stage and subjected to WISH against *ctsl1b*, *dlx3* and *ntl* to analyse convergence and extension as described in figure 4. Representative photographs are shown in (*a*) and (*c*), and quantitative analyses are shown in (*b*) and (*d*) (*n* = 3, \*\**p* < 0.01, \*\*\**p* < 0.001).

increase in convergent extension defects (electronic supplementary material, figure S13). This suggests that Wnt5b and Fak1a might not operate through the same genetic pathway to mediate convergent extension.

To further elucidate the plausible interaction between Wnt5b and Fak1a, we examined *wnt5b* and *fak1a* expressions in embryos injected with the *fak1a* or *wnt5b* MO. At the mRNA level, *fak1a* expression was significantly reduced in *wnt5b* morphants. By contrast, *wnt5b* expression was increased in *fak1a* morphants (figure 7*d*). Both immunoblotting (figure 7*e*) and IHC analyses (electronic supplementary material, figure S14) showed downregulation of the Fak1a protein in *wnt5b* morphants. Unfortunately, we were unable to examine the protein level of Wnt5b due to the lack of a specific antibody.

Furthermore, both *fak1a* and *wnt5b* mRNAs could reciprocally rescue the convergence defects of their respective morphants (figure 8) and *wnt5b* mRNA could partially rescue epiboly arrest of *fak1a* morphants (electronic supplementary material, figure S15). *wnt5b* morphants showed no significant extension defect at the dosage tested (7.5 ng MO per embryo). By contrast, *fak1a* morphants had extension defects, but they could not be rescued by co-injecting *wnt5b* mRNA.

Collectively, the lack of synergy between Fak1a and Wnt5b suggests that they might not work via the same genetic pathway. However, the cross-rescue effect between them implies that they might converge at downstream factors to mediate gastrulation.

## 3.10. Wnt5b integrates Fak1a to mediate gastrulation via modulating Rac1 and Cdc42

Both FAK and Wnt signalling pathways are well known for their roles in cytoskeleton regulation. RhoA, Cdc42 and Rac1

are key mediators for actin dynamics during gastrulation in zebrafish [39,52–55]. Thus, we hypothesized that Wnt5b and Fak1a signalling may converge to modulate small GTPase activities during gastrulation. Due to the possible antagonism between RhoA and Rac1 in coordinating F-actin organization [49,50], here we only tested whether *rac1* or *cdc42* mRNA could rescue *wnt5b* and *fak1a* morphant phenotypes. *rac1* and *cdc42* independently rescued both *fak1a* and *wnt5b* morphant phenotypes at certain dosages (electronic supplementary material, figure S16). Co-injection of 2.5 pg per embryo *rac1* or *cdc42* mRNA restored the percentage of normal embryos to about 30% and 45% in *fak* morphants, respectively (electronic supplementary material, figure S16B,D). Similarly, co-injection of 10 pg *rac1* or 2.5 pg *cdc42* mRNA per embryo also increased the percentage of normal embryos to about 40% and 55% in *wnt5b* morphants, respectively (electronic supplementary material, figure S16A,C). However, the rescue was not complete nor dose-dependent. It indicated that a good balance of different small GTPase activities may be necessary for gastrulation. Therefore, we injected subthreshold levels of *rac1* and *cdc42* mRNA to examine a possible synergistic rescue effect in both *fak1a* and *wnt5b* morphants. The injection of at 1.25 pg *rac1* or *cdc42* mRNA by themselves had minimum effects on gastrulation in WT embryos (electronic supplementary material, figure S16E) and had limited rescue effects in *fak1a* or *wnt5b* morphants. However, we observed synergistic rescue with the combination of *rac1* and *cdc42* to restore the percentage of normal embryos of *wnt5b* morphants from 22 to about 50%, which was notably higher than those of *rac1*-injected (35%) or *cdc42*-injected (38%) embryos (figure 9*a*; see a group photo in electronic supplementary material, figure S17A). An even more significant synergistic rescue was observed in *fak1a* morphants, from 7 to 45% of normal embryos

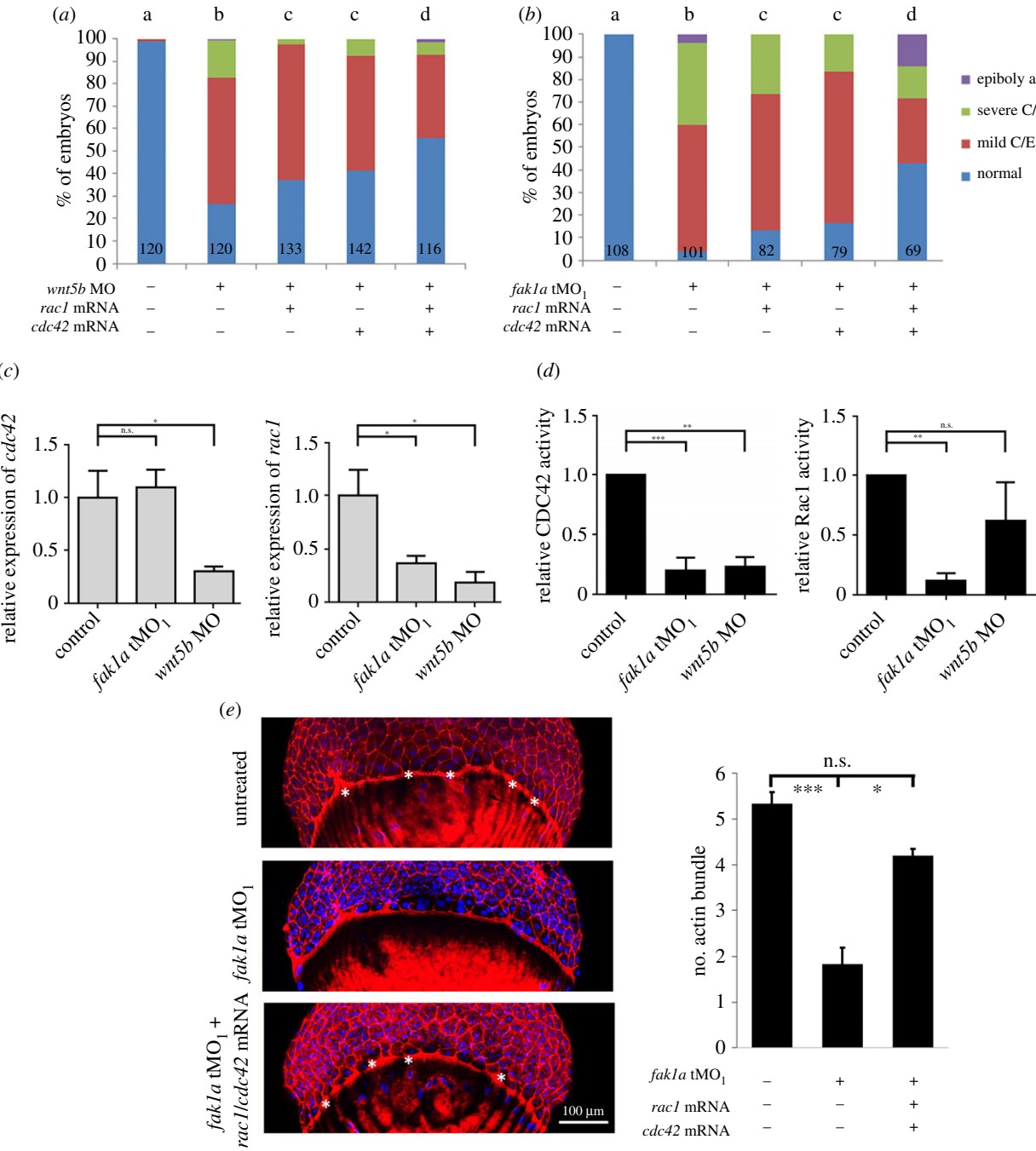

**Figure 9.** Wnt5b and Fak1a modulate Rac1 and Cdc42 to control cell migration during gastrulation. (*a,b*) Embryos were untreated or injected with 7.5 ng of the *wnt5b* MO, 5 ng of *fak1a* or 1.25 pg of *rac1* and *cdc42* mRNA. Embryos were cultured to the bud stage and classified into normal, mild/severe convergent extension (mild/severe C/E) defects, and epiboly arrest categories. The percentages of embryos in different categories are shown. Numbers of embryos observed are given at the bottom of each bar. $n = 3$. Values between groups with a significant difference ($p < 0.05$) are denoted by different letters. (*c*) Embryos injected with the *wnt5b* or *fak1a* MO were collected at the bud stage to measure the expression of *cdc42* or *rac1* by a qPCR, respectively. Ef1α served as an internal control. (*d*) Embryos were treated as in (*c*) and lysed to measure the activities of Cdc42 and Rac1 by an ELISA activity assay ($n = 3$, *$p < 0.05$). (*e*) Embryos were treated with *fak1a* MO with or without *rac1* and *cdc42* mRNA, stained and examined as described in figure 3*d*. The actin bundles between the actin ring and cap are indicated by asterisks. The numbers of actin bundles in each treatment were presented in the right bar graph ($n = 3$, *$p < 0.05$, ***$p < 0.001$).

in double rescue compared to 11 and 14% in embryos co-injected with *rac1* or *cdc42* mRNA only (figure 9*b*; see a group photo in electronic supplementary material, figure S17A). To better understand whether expression or activity level is perturbed in *fak1a* and *wnt5b* morphants, we performed qPCR and activity assays on both morphants. The expression level of *cdc42* was significantly reduced in *wnt5b* morphants but not in *fak1a* morphants. However, expression levels of *rac1* were reduced in both *fak1a* and *wnt5b* morphants (figure 9*c*). We next measured the activities of Cdc42 and Rac1 in *fak1a* and *wnt5b* morphants by detecting the GTP-bound form of small GTPase. The Cdc42 activity was significantly lower in both *fak1a* and *wnt5b* morphants. By contrast, although Rac1 activity was also lower in both *fak1a* and *wnt5b* morphants, the reduction in *wnt5b* morphants was not significant (figure 9*d*). Collectively, these results suggested that both Cdc42 and Rac1 act downstream of Wnt5b and Fak1a during gastrulation. Lastly, we tested whether *rac1* and *cdc42* mRNA could rescue the loss of F-actin bundles, and found that the co-injection of rac1 and cdc42 mRNA significantly restored the number of actin bundles in *fak1a* morphants (figure 9*e*).

royalsocietypublishing.org/journal/rsob    Open Biol. **10**: 190273

### 3.11. CRISPR/Cas9-mediated fak1a knockout causes a compensatory rise in wnt5b expression

To examine the function of *fak1a in vivo*, we generated *fak1a* mutants by a clustering of regularly interspaced short palindromic repeats (CRISPR)/Cas9-mediated technique [56]. We designed two gRNAs targeting exon 3 and one gRNA targeting exon 10 of *fak1a* (see sequences in electronic supplementary material, table S2). Genotyping analysis revealed that the one targeting exon 10 (*fak1a* E10-1) successfully caused the insertion or deletion (indels) in F0 embryos (figure 10*a*). We obtained seven deletion and four insertion mutant alleles as shown in electronic supplementary material, table S3. Among F2 mutant alleles, we validated a 5 bp deletion mutant allele, *fak1a* Δ5, by Sanger sequencing (figure 10*b*). The *fak1a* Δ5 mutant allele has an early stop codon just before the PAM site. The sense strand of the *fak1a* Δ5 allele is shown in figure 10*a* (lower strand). The translated sequence of *fak1a* Δ5 allele encodes a truncated protein of 274 amino acids without a kinase domain, a p130CAS (CAS)-binding site, a CRAF-binding site or a focal adhesion-targeting (FAT) domain (figure 10*c*).

A 5 bp deletion of the *fak1a* Δ5 allele results in the deletion of an Hae III restriction site, -GGCC-, that allowed us to separate fish with different *fak1a* genomic backgrounds (WT, heterozygous and homozygous) with an Hae III restriction digestion assay as indicated in figure 10*a*. In WT embryos (+/+), the 513 bp band (asterisk) was cleaved by Hae III into two bands of 321 and 193 bp (arrows). In heterozygous mutants (+/−), weaker 321 and 193 bp bands were present due to digestion by Hae III, and the 513 bp band remained intact because of the loss of the Hae III site in the mutated strands. In clear contrast, homozygous mutant (−/−) alleles showed only the uncut 513 bp band due to the lack of an Hae III site (figure 10*d*). *fak1a* Δ5 F2 homozygous mutants were then identified and incrossed to produce maternal zygotic (MZ) mutants, which contained a null allele as evidenced by the absence of Fak1a protein by immunoblotting (figure 10*e*).

Unexpectedly, only a portion (25%) of these embryos showed mild gastrulation defects at the bud stage but eventually grew normally at later stages (electronic supplementary material, figure S18). Similar results could be seen with two other mutant alleles, *fak1a* Δ5b and *fak1a* Δ7 (electronic supplementary material, figure S18A). *fak1a* Δ5 mutant embryos tended to have divergent and larger convergent angles but shorter extension angles compared to WT embryos. *fak1a* tMO$_1$ morphants appeared to have larger convergent and shorter extension angles compared to *fak1a* Δ5 mutant embryos. However, the effects of *fak1a* tMO$_1$ were alleviated when injected into *fak1a* Δ5 mutant embryos (figure 10*f*). These results suggest that *fak1a* Δ5 mutant embryos have notably reduced sensitivity to the *fak1a* tMO$_1$. It further demonstrates the specificity of the *fak1a* MO. Moreover, the injection of *cas9* mRNA and *fak1a* gRNA into F0 embryos also resulted in gastrulation defects in about 50% of embryos, which could be rescued by co-injecting *fak1a* mRNA (electronic supplementary material, figure S19). Together, these data suggest that gastrulation defects observed in different contexts are all due to the specific loss of *fak1a*.

The lack of severe gastrulation defects observed in *fak1a* mutants might have resulted from gene complementation and/or compensation. One of the possible candidate genes,

which might exert a complementation effect on *fak1a*, is its close relative, *fak1b* [34]. Because of the possibility that the expression of *fak1b* might complement the loss of Fak1a for gastrulation cell movements, we injected the *fak1b* MO into *fak1a* Δ5 mutant embryos but observed no further deterioration of gastrulation defects compared to WT embryos injected with the *fak1b* MO (electronic supplementary material, figure S20). We further examined changes in the gene expression of *fak1b* in WT embryos, morphants and mutants and found no significant change until 9 hpf, in which *fak1b* expression increased in both *fak1a* tMO$_1$ morphants and *fak1a* Δ5 mutants (electronic supplementary material, figure S21A). This suggests that the increase in *fak1b* expression might not explain the phenotypic difference observed between *fak1a* morphants and mutants.

Lastly, we injected a subthreshold dosage of the *wnt5b* MO at 1.875 ng, which by itself caused mild gastrulation defects in a small percentage of WT embryos, into *fak1a* Δ5 mutants and found a significant increase in the severe convergent extension defect and even epiboly arrest. The defects were only slightly higher when injected with 3.75 ng of the *wnt5b* MO (figure 10*g*). This indicates that an increase in *wnt5b* expression might occur to compensate for the loss of *fak1a* in *fak1a* Δ5 mutants.

## 4. Discussion

Despite its necessity, how FAK functions during early embryogenesis remains poorly understood. Here, we demonstrate that zebrafish Fak1a is chemically conserved with mammalian FAK and functions cooperatively with Wnt5b to regulate Rac1 and Cdc42 activities for the modulation of actin dynamics during gastrulation cell movements.

Embryos devoid of Fak1a showed abnormal epibolic movement and convergent extension. It is known that the YSL is crucial for the initiation and process of epiboly in teleosts [57,58]. By coordinating through microtubule arrays, the YSL pulls the tightly associated EVL towards the vegetal pore during epiboly [59]. In addition, actin dynamics are critical for epibolic movement. A punctuate actin band within the YSL also contributes to the progression of epiboly [46,60]. The activation of FAK can trigger F-actin remodelling and repositioning during cell migration [61]. Here, we observed that the F-actin network is severely distorted in *fak1a* morphants by an abnormal, uneven distribution of yolk syncytial nuclei and epiboly defects. By contrast, we found no notable differences in the perinuclear microtubule network or longitudinal microtubule arrays within the yolk syncytial layer between control embryos and *fak1a* morphants (see electronic supplementary material, figure S22). This further suggests an essential role of Fak1a in mediating actin dynamics, organization and YSL behaviour for gastrulation cell movements.

The deep cells of *fak1a* morphants showed significant retarded cell migration and reduced numbers of protrusion that have also been reported in mammalian cells [8,62]. These abnormalities might be due to the disturbance of FAK downstream signalling. One of the FAK downstream signalling molecules is mitogen-activated protein kinase (MAPK). Zebrafish defective in p38 or its downstream effector, MAPK-activated protein kinase 2 (MAPKAPK2), also have a disorganized F-actin network and epiboly defects [63]. Biochemically, we provide evidence to show that the reduced phosphorylation of p38, paxillin and Akt occurs in *fak1a*

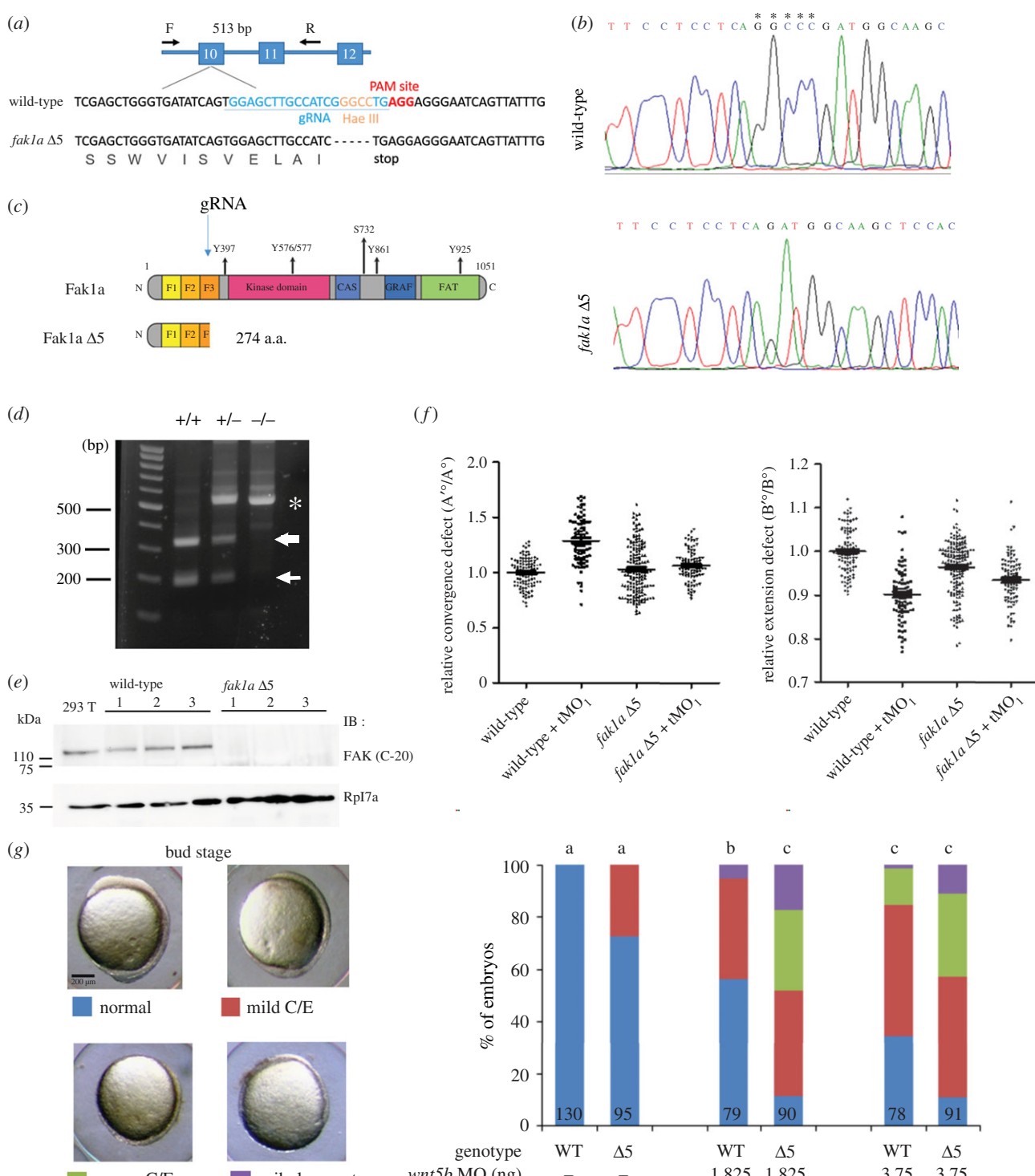

**Figure 10.** CRISPR/Cas9-mediated deletion of Fak1a results in mild gastrulation defects due to compensatory *wnt5b* expression. (a) The *fak1a* gene structure is shown with indicated exons in blue boxes. The target sites of a primer pair (F, forward primer; R, reverse primer) for amplifying a mutation detection 513-bp PCR fragment are indicated by arrows. The sense strand of wild-type *fak1a* is shown (upper strand). Exon 10 contains a PAM site shown in red. A gRNA target site of the exon 10 PAM site is labelled in blue. The Hae III site is shown in orange. The resulting sense strand of the *fak1a* Δ5 allele is shown (lower strand) with the translated amino acid in grey. (b) Partial chromatograms are shown for the wild-type and *fak1a* Δ5 alleles. Deleted nucleotides are indicated by asterisks. (c) Illustration of wild-type FAK1a and Fak1a Δ5 mutant proteins. The gRNA target site is indicated by an arrow. The mutant protein contains F1, F2 and a partial F3 domain that encodes 274 amino acids. (d) Hae III restriction digestion analysis. Genomic DNAs from tail fins of wild-type (+/+), heterozygous (+/−) or homozygous (−/−) *fak1a* were isolated and amplified by PCR using the *fak1a* forward and reverse primers indicated in A. The amplicons were digested, run in agarose gels and stained. A representative gel image is shown indicating selective molecular weight markers and sources of genomic DNAs. (e) Three different batches of wild-type and *fak1a* Δ5 mutant embryos were lysed and subjected to immunoblotting against a Fak c-terminal antibody. 293T cell lysate was used as a positive control, and zebrafish Rpl7a was an internal control. (f) Embryos were subjected to WISH against *ctsl1b/dlx3* and *ctsl1b/ntl* to analyse convergence and extension as described in figure 4. In the scatterplot, each dot represents relative convergence, and the extension defect of each embryo underwent different treatments ($n = 4$). (g) Wild-type embryos and *fak1a* Δ5 MZ mutant embryos were injected with or without a designated subthreshold amount of the *wnt5b* MO, cultured to the bud stage, and the resultant gastrulation defects were classified into normal, mild convergence and extension (C/E) defects, severe C/E defects and epiboly arrest as shown in representative photographs (side view, dorsal up, anterior to the left). The calculated percentages of embryos are shown for each class. The numbers of embryos observed are given at the bottom of each bar. $n = 3$. Values between groups with a significant difference ($p < 0.05$) are denoted by different letters.

morphants, which are all pivotal factors controlling cell mobility and migration [39,64]. Collectively, Fak1a signalling is mechanistically conserved in mediating gastrulation cell migration in zebrafish.

Our MO-based loss-of-function assay clearly demonstrated the necessity of *fak1a* for gastrulation cell movements. However, only mild gastrulation defects were observed in *fak1a* MZ mutants. This suggests that possible secondary effects might occur in *fak1a* morphants. We performed the necessary MO-specificity control experiments, including using two MOs targeting non-overlapping sites, RNA rescue and co-injection of a *p53* MO (data not shown), and all data supported the specific activity of the *fak1a* MO. A partial gastrulation defect was seen by overexpressing the dominant-negative *frnk*. In addition, *cas9* mRNA and *fak1a* gRNA induced gastrulation defects, which were also partially rescued by *fak1a* mRNA. The injection of *fak1a* mRNA also caused mild gastrulation defects. Furthermore, the convergent extension defects induced by the *fak1a* tMO₁ were notably mild in *fak1a* mutant embryos compared to WT embryos, further strengthening the specificity of tMO₁-induced gastrulation defects. These results support that perturbation of *fak1a* indeed interferes with gastrulation.

The lack of a phenotype is often observed in chemical-induced zebrafish mutants [65]. Gene compensation reported in CRISPR/Cas9-induced mutants suggests it may be a common feature in zebrafish gene knockouts [66–69]. By contrast, genetic compensation does not occur in morphants, so the corresponding phenotypes can be observed. If one only trusts the phenotypes observed in genetic mutants, the important and valid phenotypes seen in the MO-based experiments may be overlooked. To settle this issue, guidelines for the use of MO in zebrafish were reported in *PLoS Genetics* [70] authored by a group of leading zebrafish experts. Our work exactly followed the guidelines to observe the lack of response for tMO1 in a *fak1a* null MZ mutant and the identification of the compensatory factor Wnt5b. We found that the injection of *wnt5b* MO at a subthreshold level recapitulated gastrulation defects in *fak1a* Δ5 mutants, suggesting that a compensatory increase in *wnt5b* may mask gastrulation phenotypes in *fak1a* Δ5 mutants. This further supports a possible integration of Fak1a and Wnt5b signalling; however, we cannot exclude the involvement of other compensatory genes. A large-scale transcriptomic analysis would be required to unravel the causative gene(s).

The non-canonical Wnt5b pathway is known to modulate cell movements during embryogenesis [29,31]. Wnt5b may mediate the binding of Dishevelled and adenomatous polyposis coli (APC) to focal contacts to activate FAK at the leading edge of cells during migration [31,71]. We observed that *fak1a* mRNA rescued the *wnt5b* morphant phenotype and the expression, protein level and phosphorylation of C-terminal tyrosine were all downregulated in *wnt5b* morphants, suggesting that Wnt5b may act upstream of Fak1a. Interestingly, we also discovered that *wnt5b* expression increased in *fak1a* morphants, suggesting that the loss of *fak1a* might induce the compensatory expression of *wnt5b* to replenish reduced downstream signalling. However, no synergistic effect on gastrulation was observed in *fak1a* and *wnt5b* double-knockdown morphants, revealing that Fak1a and Wnt5b may not work via the same signalling pathway as previously described. Instead, Wnt5b and Fak1a might act in parallel and converging at a common downstream signalling pathway, such as small GTPases.

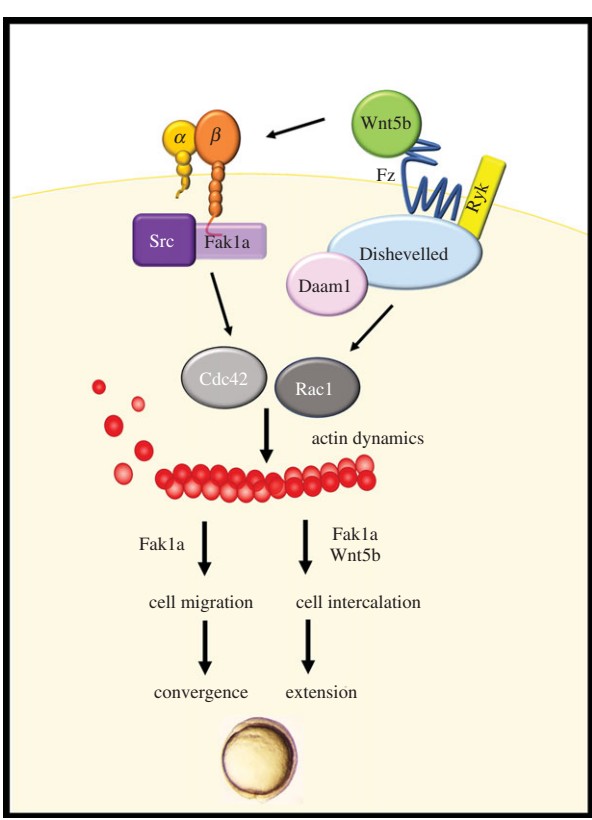

**Figure 11.** Wnt5b integrates with Fak1a to mediate cell movements during gastrulation. Gastrulation cell migration is controlled by the Wnt5b pathway to activate calcium and Dsh/Daam1. Herein, we identified that Wnt5b can also integrate with FAK signalling, and both Wnt5b and Fak1a then activate small GTPase Rac1 and Cdc42 to mediate actin dynamics during gastrulation. Convergent is mainly mediated by Fak1a-mediated cell migration and extension is exerted by cell intercalation, which requires both Fak1a and Wnt5b.

The small GTPases Rac1 and Cdc42 are known downstream effectors of both FAK and Wnt5 signalling [72–75]. Wnt5b is involved in non-canonical PCP and the Wnt/calcium pathway. An activated receptor recruits dishevelled and forms a complex with Daam1. The complex then binds with Rac1 to regulate actin dynamics. Wnt5b also regulates calcium release from the endoplasmic reticulum, and the increased intracellular calcium and diacylglycerol can activate Cdc42 through protein kinase C [76]. On the other hand, FAK can directly bind to p130cas to coordinate the activation of RAC1 [77]. FAK can also influence the functions of Cdc42 through the binding and phosphorylation of the Cdc42 effector Wiskott–Aldrich syndrome protein, N-WASP [78]. *wnt5b* is expressed at the margin of the EVL [79], and *fak1a* is ubiquitously expressed in the early embryo stage. It is possible that Wnt5b may integrate with the FAK signal to mediate the actin cytoskeleton and promote cell migration in a co-localized region. We discovered lower expression levels and activities of Rac1 and Cdc42 in both *wnt5b* and *fak1a* morphants, suggesting that they may thus relay signals from both FAK and Wnt5b to modulate actin dynamics during gastrulation. This was evidenced by the synergistic rescue of both *fak1a* and *wnt5b* morphants by Rac1 and Cdc42. Herein, we demonstrate for the first time the missing functional link between Wnt and FAK signalling to mediate gastrulation cell movements (epiboly and convergence) via the precise regulation of Rac1 and Cdc42 activities and subsequent actin dynamics.

The convergence of Wnt and FAK signalling suggests that they may collaborate to regulate the same cellular processes

royalsocietypublishing.org/journal/rsob    Open Biol. **10**: 190273

such as convergence and extension during gastrulation. Convergence and extension were once thought to be bundled processes driven by cellular intercalation [4]. Analysis of different gastrulation mutants later revealed that convergence and extension can be linked or separately regulated. Convergence and extension are both affected in *knypek/glypican6* [80] and *trilobite/strabismus* [81,82] mutants. By contrast, convergence and extension is inhibited in *bmp* [83] and *has2* [84] mutants, respectively. Convergence relies on the active cell migration of individual or groups of cells without cell rearrangement, while extension depends on cell intercalation [84]. Our data support this notion by showing that FAK affected both convergence and extension, while Wnt5b influenced only convergence. This further strengthens the idea that Wnt5b only affects cell migration, while Fak1a affects both migration and intercalation, despite their regulation of common GTPase targets.

We propose a working model to show the interaction of Wnt5b and Fak1a during gastrulation (figure 11). Conventionally, Wnt5b works via Dishevelled/Daam1 or calcium (not shown) to activate Rac1/Cdc42 and subsequent actin-dependent cell migration during gastrulation. Here, we further demonstrate that Wnt5b can work in parallel with Fak1a or activate Fak1a signalling to modulate Rac/Cadc42 and downstream signalling. During gastrulation, convergence is mainly regulated by Fak1a-mediated cell migration. On the other hand, extension, which requires cell intercalation, is mediated by Wnt5b and Fak1a signalling. In the Fak1a mutants, the inhibition of gastrulation should occur like that of *fak1a* morphants because of the reduction in Rac1 and Cdc42 activities.

However, only a mild gastrulation defect was observed in mutants due to compensatory increases in *wnt5b* expression and presumably elevated Rac1 and Cdc42 activities via a Fak1a-independent conventional pathway. Overall, we provide strong evidence to support that Wnt integrates with FAK to fine-tune Rac1 and Cdc42 activities for the differential regulation of convergence extension during gastrulation.

**Ethics.** All animal handling procedures were approved by the use of laboratory animal committee at National Taiwan University, Taipei, Taiwan (IACUC Approval ID: 97 Animal Use document No. 55).

**Data accessibility.** This article has no additional data.

**Authors' contributions.** All authors except Y.-L.T. conceived and designed the experiments. I.-C.H., T.-M.C., J.-P.L. and Y.-L.T performed experiments and analysed data. I.-C.H., T.-M.C., J.-P.L., T.-L.S and S.-J.L. wrote the manuscript. T.-L.S. and S.-J.L. directed the project. All authors read and edited the manuscript.

**Competing interests.** We declare we have no competing interests.

**Funding.** This work was supported by the Ministry of Science and Technology, Taiwan (grant no. NSC-98-2311-B-002-006-MY3 to S.-J.L. and grant nos NSC-96-2311-B-002-023-MY3 and NSC-99-2320-B-002-079-MY3 to T.-L.S.) and National Taiwan University (grant nos NTU CESRP-10R70602A5 and NTU ERP-10R80600 to S.-J.L and Frontier and Long-Term Research Grant no. 10R70821 to T.-L.S.).

**Acknowledgements.** We thank Taiwan Zebrafish core facility for providing technical guidance and plasmids. We thank Ms Yi-Chun Chuang in Technology Commons in National Taiwan University for excellent technical assistance with confocal microscopy. The authors would also like to express great appreciation to the staffs of the zebrafish Core at National Taiwan University for providing assistance in fish maintenance. We also appreciate the generous supply of CRISPR-related plasmids and protocols from Alex Schier (Harvard University).

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
