## [Reviewer comments · Open Biology]

Review History

Decision letter (RSOB-19-0265.R0)

11-Nov-2019

Dear Professor Lee,

We are writing to inform you that your manuscript RSOB-19-0265 entitled "Wnt5b integrates with Fak1a to mediate gastrulation cell movements via Rac1 and Cdc42" has, in its current form, been rejected for publication in Open Biology.

Please find below the comments made by the referees, not including confidential reports to the Editor, which I hope you will find useful. If you do choose to resubmit your manuscript, please upload a 'response to referees' document including details of how you have responded to the comments, and the adjustments you have made.

The resubmission will be treated as a new manuscript and will re-enter the review process. Every attempt will be made to use the original referees, but this cannot be guaranteed. Please note that resubmissions must be submitted within six months of the date of this email. In exceptional circumstances, extensions may be possible if agreed with the Editorial Office. Manuscripts submitted after this date will be automatically rejected.

To upload a resubmitted manuscript, log into <http://mc.manuscriptcentral.com/rsob> and enter your Author Centre, where you will find your manuscript title listed under "Manuscripts with Decisions." Under "Actions," click on "Create a Resubmission." Please be sure to indicate in your cover letter that it is a resubmission, and supply the previous reference number.

Sincerely,
The Open Biology Team
mailto: openbiology@royalsociety.org

Comments:

This manuscript was a resubmission of the previously reviewed manuscript RSOB-17-0043. The previous manuscript was reviewed by two reviewers and resubmission was invited. However, this resubmitted manuscript does not tell what improvements have been made and point-to-point responses to the reviewers' comments are missing. The authors need to provide these information before further consideration.

Author's Response to Decision Letter for (RSOB-19-0265.R0)

See Appendix A.

RSOB-19-0273.R0

Review form: Reviewer 1

Recommendation

Accept with minor revision (please list in comments)

Do you have any ethical concerns with this paper?

No

Comments to the Author

The authors have addressed my original concerns by conducting a substantial amount of additional experiments and changing the text of the manuscript where appropriate. My only suggestion at this point is that, as fak1a knockout may cause a compensatory rise in wnt5b expression, whether an increase in wnt5b expression can be found in fak1a Δ 5 mutants?

Review form: Reviewer 2

Recommendation

Accept with minor revision (please list in comments)

Do you have any ethical concerns with this paper?

No

Comments to the Author

This revised manuscript has improved and some of my concerns have been resolved properly. However, my major concern about the specificity of phenotypes displayed in the *fak1a* morphants is still not addressed adequately.

In the text, they described that injection of the *fak1b* MO into *fak1a* $\Delta 5$ mutant embryos, but observed no further deterioration of gastrulation defects compared to WT embryos injected with the *fak1b* MO. In addition, they noticed *fak1b* expression in WT, morphants and mutants is not changed until 9 hpf, but is increased in both the morphants and mutants after 9hpf. These observations clearly suggest the *fak1b* expression cannot complement the loss of *fak1a*. However, in order to explain this inconsistency between the morphants and the mutants, the authors examined *wnt5b* expression and thought that an increase in *wnt5b* expression might occur to compensate for the loss of *fak1a* in the mutants. *Fak1a* and *wnt5b* are two different genes. It is hard to accept that *wnt5b* compensation can explain the inconsistency between the *fak1a* morphants and the *fak1a* mutants. In order to get this manuscript to be published, the authors should carefully explain and discuss their observations.

Decision letter (RSOB-19-0273.R0)

19-Dec-2019

Dear Professor Lee,

We are pleased to inform you that your manuscript RSOB-19-0273 entitled "Wnt5b integrates with Fak1a to mediate gastrulation cell movements via Rac1 and Cdc42" has been accepted by the Editor for publication in Open Biology. The reviewer(s) have recommended publication, but also suggest some minor revisions to your manuscript. Therefore, we invite you to respond to the reviewer(s)' comments and revise your manuscript.

Please submit the revised version of your manuscript within 7 days. If you do not think you will be able to meet this date please let us know immediately and we can extend this deadline for you.

- 1) A text file of the manuscript (doc, txt, rtf or tex), including the references, tables (including captions) and figure captions. Please remove any tracked changes from the text before submission. PDF files are not an accepted format for the "Main Document".

2) A separate electronic file of each figure (tiff, EPS or print-quality PDF preferred). The format should be produced directly from original creation package, or original software format. Please note that PowerPoint files are not accepted.

3) Electronic supplementary material: this should be contained in a separate file from the main text and meet our ESM criteria (see <http://royalsocietypublishing.org/instructions-authors#question5>). All supplementary materials accompanying an accepted article will be treated as in their final form. They will be published alongside the paper on the journal website and posted on the online figshare repository. Files on figshare will be made available approximately one week before the accompanying article so that the supplementary material can be attributed a unique DOI.

Online supplementary material will also carry the title and description provided during submission, so please ensure these are accurate and informative. Note that the Royal Society will not edit or typeset supplementary material and it will be hosted as provided. Please ensure that the supplementary material includes the paper details (authors, title, journal name, article DOI). Your article DOI will be 10.1098/rsob.2016[*last 4 digits of e.g. 10.1098/rsob.20160049*].

4) A media summary: a short non-technical summary (up to 100 words) of the key findings/importance of your manuscript. Please try to write in simple English, avoid jargon, explain the importance of the topic, outline the main implications and describe why this topic is newsworthy.

Images

Data-Sharing

It is a condition of publication that data supporting your paper are made available. Data should be made available either in the electronic supplementary material or through an appropriate repository. Details of how to access data should be included in your paper. Please see <http://royalsocietypublishing.org/site/authors/policy.xhtml#question6> for more details.

Data accessibility section

Sincerely,

The Open Biology Team

<mailto:openbiology@royalsociety.org>

Reviewer(s)' Comments to Author:

Referee: 1

Comments to the Author(s)

The authors have addressed my original concerns by conducting a substantial amount of additional experiments and changing the text of the manuscript where appropriate. My only

suggestion at this point is that, as fak1a knockout may cause a compensatory rise in wnt5b expression, whether an increase in wnt5b expression can be found in fak1a Δ 5 mutants?

Referee: 2

Comments to the Author(s)

This revised manuscript has improved and some of my concerns have been resolved properly. However, my major concern about the specificity of phenotypes displayed in the fak1a morphants is still not addressed adequately.

In the text, they described that injection of the fak1b MO into fak1a Δ 5 mutant embryos, but observed no further deterioration of gastrulation defects compared to WT embryos injected with the fak1b MO. In addition, they noticed fak1b expression in WT, morphants and mutants is not changed until 9 hpf, but is increased in both the morphants and mutants after 9hpf. These observations clearly suggest the fak1b expression cannot complement the loss of fak1a. However, in order to explain this inconsistency between the morphants and the mutants, the authors examined wnt5b expression and thought that an increase in wnt5b expression might occur to compensate for the loss of fak1a in the mutants. Fak1a and wnt5b are two different genes. It is hard to accept that wnt5b compensation can explain the inconsistency between the fak1a morphants and the fak1a mutants. In order to get this manuscript to be published, the authors should carefully explain and discuss their observations.

Author's Response to Decision Letter for (RSOB-19-0273.R0)

See Appendix B.

Decision letter (RSOB-19-0273.R1)

27-Jan-2020

Dear Professor Lee

We are pleased to inform you that your manuscript entitled "Wnt5b integrates with Fak1a to mediate gastrulation cell movements via Rac1 and Cdc42" has been accepted by the Editor for publication in Open Biology.

Please note that the article processing charge has been waived and no fees are payable.

Sincerely,

The Open Biology Team

mailto: openbiology@royalsociety.org

Appendix A

Reviewer(s)' Comments to Author(s):

Referee: 1

Comments to the Author(s)

Hung et al., found the loss of Fak1a impairs epiboly, convergent extension and hypoblast cell migration non-cell-autonomously in zebrafish embryos using antisense morpholino (MO) and dominant-negative approaches. In addition, they found overexpression of fak1a or wnt5b mRNA could cross rescue convergence defects induced by wnt5b or fak1a MO, respectively. They showed that Wnt5b and Fak1a were converged in regulating Rac1 and Cdc42, which could synergistically rescued wnt5b and fak1a morphant phenotypes. They also generated several alleles of fak1a mutants using CRISPR/Cas9, but only mild gastrulation defects exhibited in those mutants.

In this manuscript, the most of experiments was well designed, and the writing is clear for the readers to follow. However, there are several concerns regarding the specificity of phenotypes.

Major concerns:

1. No experiments were performed to confirm the effect of morpholinos. The phenotypes caused by fak1a-MO1 and fak1a-MO2 were inconsistent. Because the phenotypes caused by fak1a-MO1 were obvious, the authors chose fak1a-MO1 to conduct subsequent experiments, which raised a concern whether the phenotypes exhibited in fak1a-MO1 was specific or non-specific. Similarly, the phenotypes in fak1a-MO is more obvious than that in fak1b-MO, so the authors chose fak1a-MO for this study.

Reply:

- The translational blocking effect of *fak1a*-tMO₁ was shown by the reduction of Fak protein level by Western analysis (Fig. 2B)
- Both *fak1a*-tMO₁ and *fak1a*-tMO₂ caused similar epiboly and convergent extension defects (see Fig. 2,4 for tMO₁ and Fig, S9 for tMO₂). Their phenotypes were consistent. For the simplicity of experiments we used a “relatively” more effective tMO₁ for subsequent experiments. To avoid confusion, we rephrased the paragraph as stated above.
- We used a control MO and a N25 random MO and they both showed no embryonic toxicity at the dosages tested. The specificity of *fak1a* MO was vigorously validated by similar phenotypes induced by two MOs (tMO₁ and tMO₂) targeting non-overlapping sites. The phenotypes induced by both *fak1a*

MOs could be significantly rescued by co-injecting *fak1a* mRNA that further demonstrated the phenotypes were caused by the loss of Fak1a.

- Similar situations as stated previously, the MO knockdown effects was similar but more severe in *fak1a* morphants compared to *fak1b* morphants. The knockdown effects of both *fak1b* MO had also been tested as that in the *fak1a* MO experiments. We chose to perform the *fak1a* experiments to better reveal the effects. In addition, the Fak1a is closer to the mammalian FAKs as shown in the phylogenetic tree (Fig, S2).

2. The major concern is the inconsistency between morphants and the mutant generated by CRISPR/Cas9. The author claimed that the protein level of *fak1a* was reduced to 30% in morphants, but undetectable in the mutant. However, the mild defects exhibited in the mutant and dramatic defects exhibited in the morphants. Even though, the authors guessed that this inconsistency might be due to redundant function of other factors, it is hard to convince this reviewer because in the homozygous of mutant the maternal effect of *fak1a* was disappeared, the phenotypes should be more obvious than that in the morphants.

Reply:

The lack of a phenotype is often observed in chemical-induced zebrafish mutants(1). Gene compensation reported in CRISPR/Cas9-induced mutants suggests it may be a common feature in zebrafish gene knockouts(2-5). On the other hand, genetic compensation does not occur in morphants, so the corresponding phenotypes can be observed. If one only trusts the phenotypes observed in genetic mutants, the important and valid phenotypes seen in the MO-based experiments may be overlooked. To settle this issue, guidelines for the use of morpholino in zebrafish were reported in PLOS Genetics (6) authored by a group of leading zebrafish experts. Our work exactly followed the guidelines to observe the lack of response for tMO₁ in a *fak1a* null maternal zygotic mutant (Fig. 10F) and the identification of a compensatory factor Wnt5b (Fig. 10G). Therefore, we are confident that the phenotypes observed in *fak1a* morphants were not nonspecific artifacts. We include the above arguments in the Discussion section.

Minor concerns

1. It is better to use the probes labeled by different colors in the same embryo for in situ hybridization assays.

Reply: Yes, it is better to use different color probes in the in situ hybridization assay. However, multi-color *in situ* can be tricky and the expression patterns of gene studied

are specific that the single color staining was sufficient for quantitative measurements presented.

2. The authors should directly provide some pictures showing the obvious different phenotypes together with the column figures.

Reply:

Most figures provided contained at least representative photo for each class except Fig. 9. So we include group photos for Fig. 9A and B in Supplementary figures.

3. In page 25, line 12, the authors indicated that fak1b-MO caused obvious defects in convergence and extension, but in page 25, line 17-19, the authors indicated that the defects exhibited in fak1b-MO is not as dramatic as that in fak1a-MO. What is the standard for phenotype analysis?

Reply:

Sorry for the confusion! We rephrase as following: “The fak1b-MO caused notable defects in convergence and extension, but was comparatively less severe than that of *fak1a*-MO.”

Referee: 2

Comments to the Author(s)

In the present manuscript the authors show Wnt5b-Fak1a pathway regulates gastrulation cell movements via Rac1 and Cdc42 during early embryogenesis. Wnt5b, Fak1, Rac1 and Cdc42 have been proved to be critical for gastrulation cell movements. The new finding in this paper is the functional interaction between Wnt5b and Fak1 signaling in coordinated cell movements. Though the authors do a good job in respect of describing a requirement of wnt5b and fak1a for cell mobility in zebrafish morphological cell movements, the conclusion that Fak1a works downstream of Wnt5b signaling in mediating gastrulation is still open to discussion.

1. In Figure 7A, It is true that fak1a overexpression can rescue the convergence and extension defects of ppt mutants, and then the authors draw a conclusion that the Fak1a may work downstream of Wnt5b signaling in mediating gastrulation. If the genetic interactions of Wnt5b-Fak1a pathway do exist, why there is no combinatory increase in defects of convergence and extension in embryos injected with both wnt5b and fak1a MOs (Supplementary Fig. S13)? Furthermore, if wnt5b

acts upstream of fak1a, why injection of wnt5b mRNA could rescue the convergence defects in fak1a morphants (Figure 8 C and D)?

Reply:

Thanks for the comments! Yes, with the data presented the genetic interaction between Wnt5b and Fak1a is weak. So this point we tend to propose that they are independent pathways, but converge on the same small GTPases, CDC42 and Rac1.

2. Based on the results of cell transplantation experiments, the authors concluded that zebrafish FAK1a mediates gastrulation cell migration in a non-cell-autonomous manner. This conclusion is confusing. As the authors suggested, Fak1a is central to the formation of F-actin network to regulate cell movements via Rac1 and Cdc42. So, at least, Fak1a plays important functions in a cell-autonomous manner.

Reply:

Sorry for the confusion! Yes, Fak1a mediates cell protrusions in a cell-autonomous manner possible via regulating actin dynamics. However, as described in the text, the effective protrusions toward dorsal and vegetal poles were reduced in a morphant host that was why we conclude FAK1a mediates gastrulation cell migration in a non-cell-autonomous manner. To avoid confusion, we rephrase the sentence as following: FAK1a mediates the direction of gastrulation cell migration in a non-cell-autonomous manner. We also change the section title to the following: “FAK1a autonomously mediates cell protrusions, but non-cell autonomously regulates the directions of cell movement during gastrulation”.

3. It is a vigorous attempt to compare the protrusion dynamics between control embryos and fak1a morphants. But the images in Figure 5 and Figure 6 for morphological analysis of cell protrusions have not enough quality to convince the reader of the author's contention.

Reply:

Sorry for the poor resolution in the images provided. The images are snap shots from respective movies. Dynamic cell protrusions can be better seen in the respective supplemental Moves S1-S8.

4. It has been reported that RhoA also acts downstream of Wnt5 to regulate convergence and extension movements in zebrafish embryos (Cell Signal 2006, 18:359-372; Cell Research 2017, 27:202-225). Does Fak1a regulate gastrulation cell movements via RhoA?

Reply:

Due to the possible antagonism between RhoA and Rac1 in coordinating F-actin

organization^{48,49}, here we only tested whether *rac1* or *cdc42* mRNA. The rationale is added to the corresponding results section.

1. Kettleborough RN, Busch-Nentwich EM, Harvey SA, Dooley CM, de Bruijn E, van Eeden F, et al. A systematic genome-wide analysis of zebrafish protein-coding gene function. *Nature*. 2013;496(7446):494-7.
2. Zhang P, Bai Y, Lu L, Li Y, Duan C. An oxygen-insensitive Hif-3alpha isoform inhibits Wnt signaling by destabilizing the nuclear beta-catenin complex. *Elife*. 2016;5.
3. Spicer OS, Wong TT, Zmora N, Zohar Y. Targeted Mutagenesis of the Hypophysiotropic Gnrh3 in Zebrafish (*Danio rerio*) Reveals No Effects on Reproductive Performance. *PloS one*. 2016;11(6):e0158141.
4. Rossi A, Kontarakis Z, Gerri C, Nolte H, Holper S, Kruger M, et al. Genetic compensation induced by deleterious mutations but not gene knockdowns. *Nature*. 2015;524(7564):230-3.
5. Lin MJ, Lee SJ. Stathmin-like 4 is critical for the maintenance of neural progenitor cells in dorsal midbrain of zebrafish larvae. *Sci Rep*. 2016;6:36188.
6. Stainier DYR, Raz E, Lawson ND, Ekker SC, Burdine RD, Eisen JS, et al. Guidelines for morpholino use in zebrafish. *PLoS Genet*. 2017;13(10):e1007000.

Appendix B

Referee: 1

Comments to the Author(s)

The authors have addressed my original concerns by conducting a substantial amount of additional experiments and changing the text of the manuscript where appropriate. My only suggestion at this point is that, as *fak1a* knockout may cause a compensatory rise in *wnt5b* expression, whether an increase in *wnt5b* expression can be found in *fak1a* $\Delta 5$ mutants?

Reply:

As demonstrated in the Supplementary Fig. S14, there is an increase in *wnt5b* expression in *fak1a* $\Delta 5$ mutants.

Referee: 2

Comments to the Author(s)

This revised manuscript has improved and some of my concerns have been resolved properly. However, my major concern about the specificity of phenotypes displayed in the *fak1a* morphants is still not addressed adequately. In the text, they described that injection of the *fak1b* MO into *fak1a* $\Delta 5$ mutant embryos, but observed no further deterioration of gastrulation defects compared to WT embryos injected with the *fak1b* MO. In addition, they noticed *fak1b* expression in WT, morphants and mutants is not changed until 9 hpf, but is increased in both the morphants and mutants after 9 hpf. These observations clearly suggest the *fak1b* expression cannot complement the loss of *fak1a*. However, in order to explain this inconsistency between the morphants and the mutants, the authors examined *wnt5b* expression and thought that an increase in *wnt5b* expression might occur to compensate for the loss of *fak1a* in the mutants. *Fak1a* and *wnt5b* are two different genes. It is hard to accept that *wnt5b* compensation can explain the inconsistency between the *fak1a* morphants and the *fak1a* mutants. In order to get this manuscript to be published, the authors should carefully explain and discuss their observations.

Reply:

Indeed, we demonstrated that *Fak1a* and *Wnt5b* work in parallel during gastrulation as stated by this reviewer. However, we also found the synergistic rescue of both *fak1a* and *wnt5b* morphants by *Rac1* and *Cdc42*. It suggests a functional link between Wnt and FAK signaling to mediate gastrulation cell movements (epiboly and convergence) via the precise regulation of *Rac1* and *Cdc42* activities and subsequent

actin dynamics. The above description has been explained in detail in the Discussion section.